# Lsm12 is an NAADP receptor and a two-pore channel regulatory protein required for calcium mobilization from acidic organelles

Jiyuan Zhang[1,3], Xin Guan[1,3], Kunal Shah[1] & Jiusheng Yan [1,2✉]

Nicotinic acid adenine dinucleotide phosphate (NAADP) is a potent $Ca^{2+}$-mobilizing second messenger which uniquely mobilizes $Ca^{2+}$ from acidic endolysosomal organelles. However, the molecular identity of the NAADP receptor remains unknown. Given the necessity of the endolysosomal two-pore channel (TPC1 or TPC2) in NAADP signaling, we performed affinity purification and quantitative proteomic analysis of the interacting proteins of NAADP and TPCs. We identified a Sm-like protein Lsm12 complexed with NAADP, TPC1, and TPC2. Lsm12 directly binds to NAADP via its Lsm domain, colocalizes with TPC2, and mediates the apparent association of NAADP to isolated TPC2 or TPC2-containing membranes. Lsm12 is essential and immediately participates in NAADP-evoked TPC activation and $Ca^{2+}$ mobilization from acidic stores. These findings reveal a putative RNA-binding protein to function as an NAADP receptor and a TPC regulatory protein and provides a molecular basis for understanding the mechanisms of NAADP signaling.

[1] Department of Anesthesiology & Perioperative Medicine, The University of Texas MD Anderson Cancer Center, Houston, TX, USA. [2] Neuroscience and Biochemistry & Cell Biology Programs, The University of Texas MD Anderson Cancer Center UT Health Graduate School of Biomedical Sciences, Houston, TX, USA. [3] These authors contributed equally: Jiyuan Zhang, Xin Guan. ✉email: jyan1@mdanderson.org

ntracellular $Ca^{2+}$ signaling, which occurs via changes or oscillation in cytosolic $Ca^{2+}$ concentration, controls almost every aspect of cellular functions and physiological processes. $Ca^{2+}$ mobilization from intracellular stores mediated by second messengers plays a critical role in the regulation of cytosolic $Ca^{2+}$ levels. Among the three known $Ca^{2+}$-mobilizing second messengers in mammalian cells, nicotinic acid adenine dinucleotide phosphate (NAADP), which differs from the enzyme cofactor nicotinamide adenine dinucleotide phosphate (NADP) by a hydroxyl group (Supplementary Fig. 1), is the most potent as it is effective in the low nanomolar range[1–3]. The other two $Ca^{2+}$-mobilizing second messenger molecules, inositol 1,4,5-trisphosphate ($IP_3$) and cyclic ADP-ribose (cADPR), are known to mobilize $Ca^{2+}$ from the endoplasmic reticulum (ER) $Ca^{2+}$ stores by activating $IP_3$ receptors and ryanodine receptors, respectively. In contrast, NAADP mobilizes $Ca^{2+}$ from acidic organelles of endosomes and lysosomes (endolysosomes) through an as yet poorly understood mechanism[4] (Supplementary Fig. 1).

NAADP signaling is broadly present in different mammalian cells, involved in many cellular and physiologic processes, and implicated in many diseases including lysosomal storage diseases, diabetes, autism, and cardiovascular, blood, and muscle diseases[4,5]. Accumulating evidence indicates that endolysosomal two-pore channels (TPCs) are necessary for NAADP-evoked $Ca^{2+}$ release. TPCs are homodimeric cation channels consisting of two functional isoforms, TPC1 and TPC2, in humans and mice. Manipulation of TPC expression in cell lines results in changes in NAADP-evoked $Ca^{2+}$ release[6–9]. Cells' response to NAADP is eliminated in TPC knockout (KO) mice[6,10,11] and rescued by reexpression of TPCs in TPC1/2 KO mice[11]. NAADP sensitivity and $Ca^{2+}$ permeability were reported for TPC currents recorded on isolated enlarged endolysosomes[12–14] and reconstituted lipid bilayers[15,16]. Thus, TPCs had been considered the most likely candidates of NAADP-responsive $Ca^{2+}$ release channels on endolysosomal membranes. However, other researchers reported that TPCs were not only fully insensitive to NAADP but also $Na^+$-selective channels with very limited $Ca^{2+}$ permeability in conventional patch-clamp recordings of exogenous mammalian TPCs expressed on whole enlarged endolysosomes[17], plasma membranes[17,18], or plant vacuoles[19,20]. Furthermore, TPCs were not labeled by a photoreactive NAADP analog[21] in spite of the reported association between NAADP and TPC-enriching protein or membrane preparations in other studies[6,22]. Thus, the molecular identity of the NAADP receptor remains elusive and the molecular basis of NAADP-evoked $Ca^{2+}$ release remains controversial and poorly understood.

## Results

### Identification of Lsm12 as an interacting partner of both NAADP and TPCs.
Given the essential role of TPCs in NAADP signaling, we hypothesize that the NAADP receptor is part of a TPC-containing NAADP signaling multiprotein complex (Supplementary Fig. 1) in which the receptor interacts with the channels either directly as an accessory protein as previously proposed[23,24] or indirectly via a different mechanism. We chose the human embryonic kidney (HEK) 293 cell line as a mammalian cell model for identification and functional characterization of the NAADP receptor because of its low endogenous TPC1 and TPC2 expression[6] and robust NAADP-evoked $Ca^{2+}$ release upon the heterologous expression of TPCs[6–9]. To identify the most likely protein candidate of the NAADP receptor, we used TPC1, TPC2, and NAADP as baits in affinity purification/precipitation and a stable isotope labeling by amino acids in cell cultures (SILAC)-based quantitative proteomic approach to identify their mutually interacting proteins (Fig. 1a). To immobilize NAADP

for affinity precipitation of its interacting proteins, we crosslinked NAADP to adipic acid dihydrazide-agarose via pyridine ribose, as described previously[25] (Fig. 1b). Upon transient expression of human TPC1 (TPC1-eGFP-FLAG) and TPC2 (TPC2-eGFP-FLAG) in HEK293 cells, we used immobilized NAADP to pull down the NAADP-interacting proteins and an anti-FLAG antibody to pull down the TPC1 and TPC2 and their interacting proteins (Fig. 1a). To obtain negative control samples, affinity precipitation in the absence of NAADP (left panel) or input samples of HEK293 cells expressing eGFP-FLAG only (right panel) were employed. Putative interacting proteins were identified by liquid chromatography-tandem mass spectrometry (LC-MS/MS) as differential proteins in the test and control samples that were differentially labeled by heavy ($^{13}C$) and light ($^{12}C$) isotopes in Arg and Lys (Fig. 1a). Among the four lists of identified putative interacting proteins, a Sm-like (Lsm) protein, Lsm12, was uniquely positioned as an interacting protein shared by TPC1, TPC2, and NAADP (both TPC1- and TPC2-expressing cells) (Fig. 1c and Supplementary Table 1). As demonstrated by the large MS peak ratios ($\geq$3) of heavy ($^{13}C_6$-Arg/Lys) and light ($^{12}C_6$-Arg/Lys) labeled peptides (Fig. 1d and Supplementary Fig. 2), Lsm12 was differentially pulled down in the test and in the negative control sample preparations in all four conditions (protein samples of TPC1-eGFP-FLAG or TPC2-eGFP-FLAG expressing cells pulled down by immobilized NAADP and by an anti-FLAG antibody). Immunoblotting confirmed the presence of Lsm12 in the TPC-interactomes, that were pulled down by an anti-FLAG antibody from TPC1-FLAG and TPC2-FLAG expressing cells (Fig. 1e); and in the NAADP-interactomes, that were pulled down by immobilized NAADP from both TPC1- and TPC2-expressing cells (Fig. 1f).

### Lsm12 is required and immediately participates in NAADP-evoked $Ca^{2+}$ release.
To determine the role of Lsm12 in NAADP signaling, we generated a HEK293 Lsm12-KO cell line by using the CRISPR/Cas9 genome editing method (Supplementary Fig. 3a). The *Lsm12* alleles in the Lsm12-KO cells have either 1 bp deletion or 68 bps insertion at the start of Lsm12's exon 3 (Supplementary Fig. 3b). Both mutations result in framing errors after the amino acid residue S46 and thus a truncation of 149 amino acid residues (195 residues in full length). Immunoblotting with an anti-Lsm12 antibody, which recognizes the C-terminal region (Supplementary Fig. 3c), showed that Lsm12 expression was undetectable in the cell lysate of Lsm12-KO cells (Fig. 2a). Immunofluorescence staining with the same anti-Lsm12 antibody showed that the staining was broadly present in the cytoplasm in WT cells but became invisible in Lsm12-KO cells (Fig. 2b). The transiently expressed exogenous recombinant Lsm12 (Myc and FLAG-tagged) replenished Lsm12 expression in Lsm12-KO cells (Fig. 2a, b).

To determine the functional role of Lsm12, we transiently expressed TPC2 and performed a $Ca^{2+}$-imaging assay in HEK293 WT and Lsm12-KO cells. We used the genetically encoded ultrasensitive fluorescent $Ca^{2+}$ sensor protein GCaMP6f[26] to monitor changes of the intracellular $Ca^{2+}$ level. Upon the microinjection of NAADP, we observed an increase in the fluorescence of the $Ca^{2+}$ sensor in an NAADP concentration-dependent manner. A bell-shaped dose-response, which was peaked at 100 nM NAADP (pipette solution), was observed when low concentrations of NAADP in pipette solution were injected (Supplementary Fig. 4a). When higher concentrations (1 and 10 μM in pipette solution) of NAADP were used, the nonspecific response was increased as some responses remained in the absence of exogenous TPC expression or in the presence of cell pretreatment with *trans*-Ned-19 (10 μM), an antagonist of

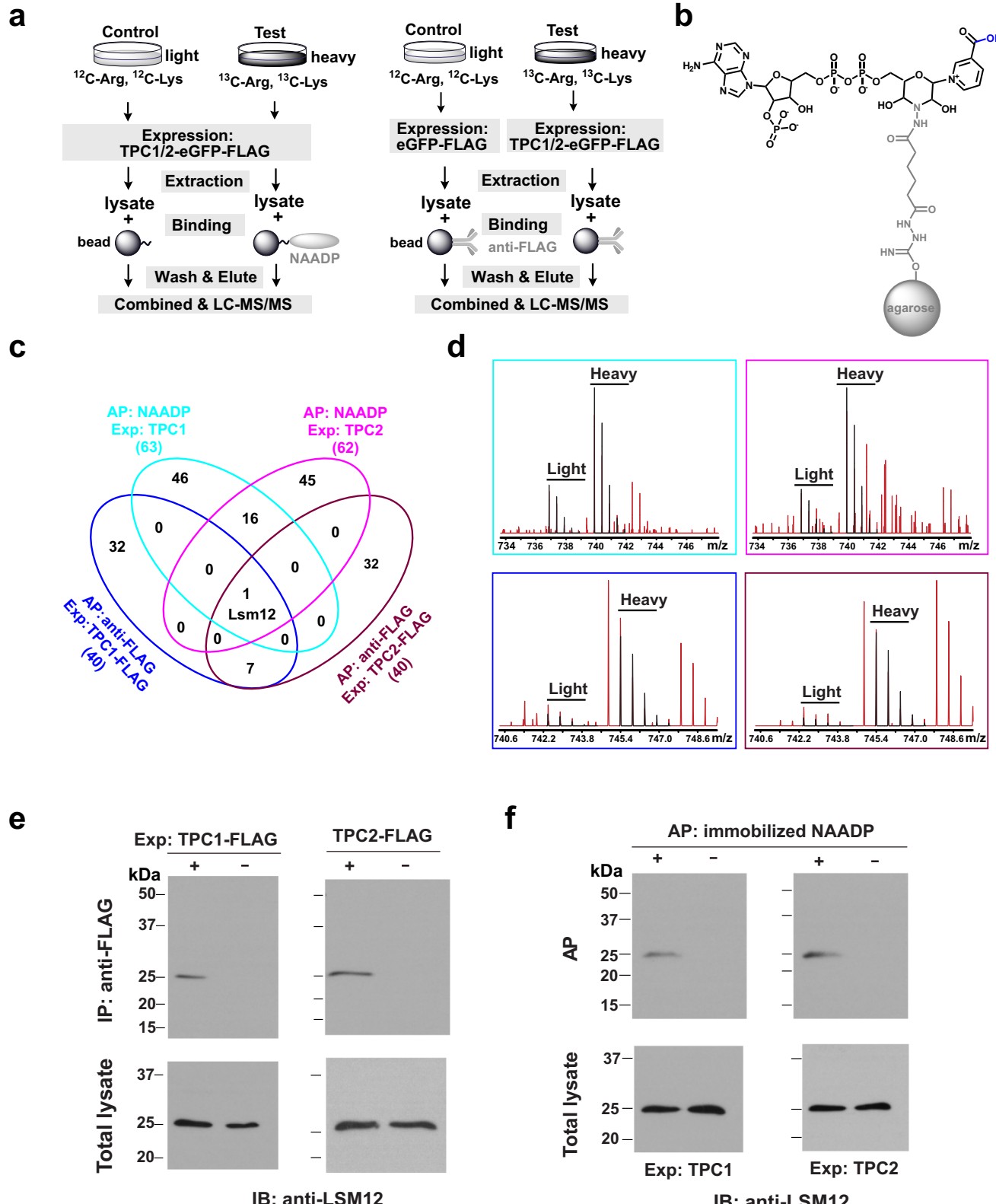

**Fig. 1 Identification of Lsm12 as a putative NAADP receptor. a** Strategies of affinity precipitation- and SILAC-based quantitative proteomic analyses of interacting proteins of NAADP (left) and TPCs (right). **b** Chemical structures of immobilized NAADP. **c** Numbers of overlapping and non-overlapping putative interacting proteins identified by MS in the indicated four types of affinity-precipitated samples. **d** Representative heavy-light peak pairs (black) of Lsm12 peptides. Peptide SQAQQPQKEAALS[194] (top) was identified in the samples affinity-purified by immobilized NAADP from TPC1- and TPC2-expressing cells. Peptide LQGEVVAFDYQSK[37] was identified in the TPC1 and TPC2 complexes affinity-purified by an anti-FLAG antibody. **e** Immunoblot of endogenous Lsm12 in samples immunoprecipitated by an anti-FLAG antibody from cells expressing FLAG-tagged TPC1 or TPC2. **f** Immunoblot of endogenous Lsm12 in samples pulled down by immobilized NAADP from TPC1- and TPC2-expressing cells. AP affinity-precipitation. Exp. transient expression, IB immunoblot.

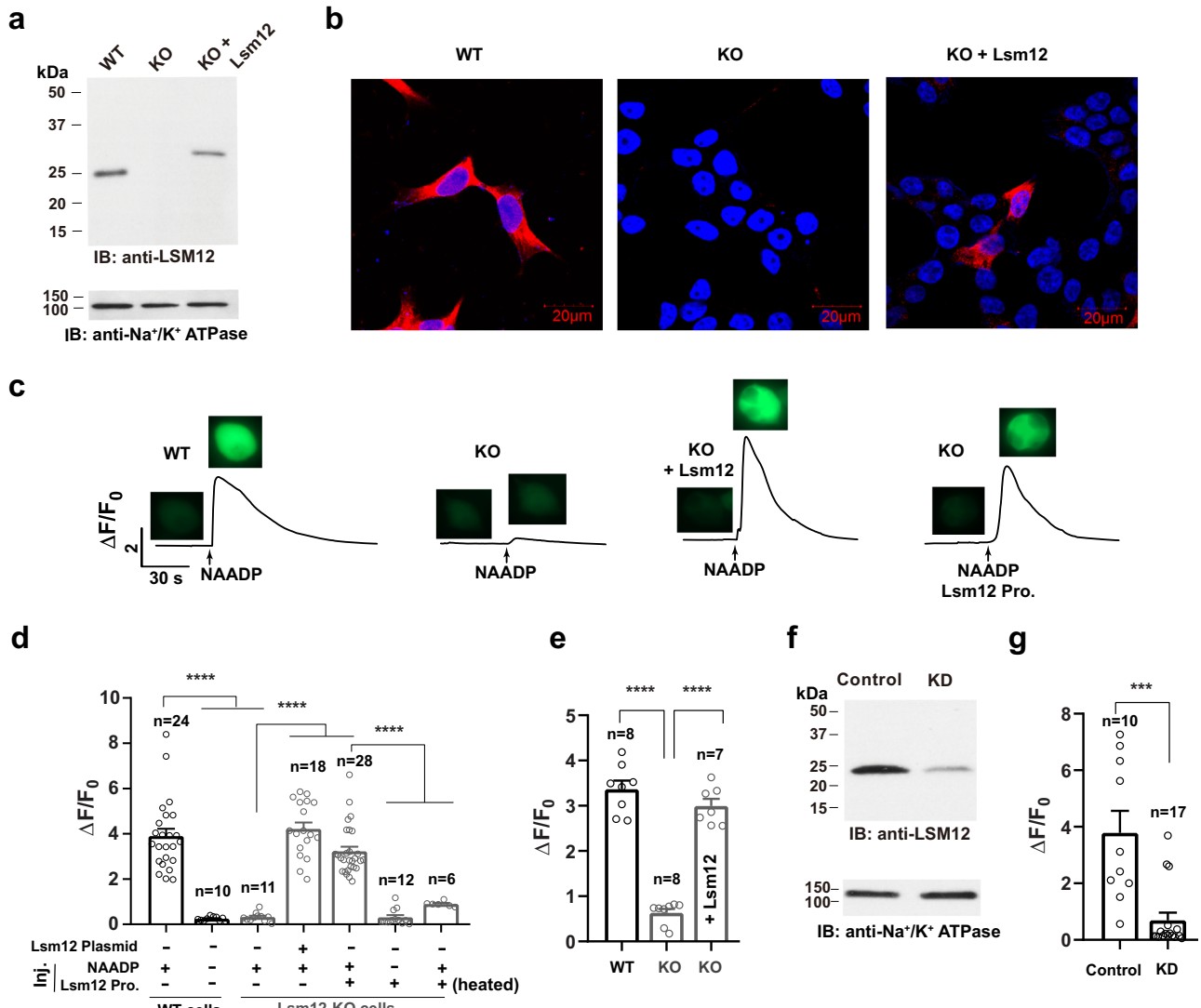

**Fig. 2 Lsm12 is essential for NAADP-evoked Ca$^{2+}$ release. a, b** Immunoblot (**a**) and immunofluorescence (**b**) assays show the loss of Lsm12 expression in the Lsm12-KO cell line. Lsm12 and cell nuclei are shown in red and blue, respectively. **c** Time course and cell images of NAADP-induced change in fluorescence of Ca$^{2+}$ indicator in TPC2-expressing HEK293 WT and Lsm12-KO cells with/without exogenous Lsm12 expression or protein injection. Cell images were taken before NAADP injection and at the peak of Ca$^{2+}$ increase after injection. **d** Averaged NAADP-induced changes in Ca$^{2+}$ indicator fluorescence in TPC2-expressing HEK293 WT and Lsm12-KO cells. Exogenous Lsm12 was introduced to cells by either plasmid transfection or protein injection (labeled at the bottom). **e** Averaged NAADP-induced changes in Ca$^{2+}$ indicator fluorescence in TPC1-expressing HEK293 WT and Lsm12-KO cells. Exogenous Lsm12 was introduced to cells by plasmid transfection (labeled in the column). **f** Immunoblot of Lsm12 in cell lysates of SK-BR-3 control and Lsm12-KD cells. **g** Averaged NAADP-induced changes in Ca$^{2+}$ indicator fluorescence in TPC2-expressing SK-BR-3 control and Lsm12-KD cells. Data were presented as mean value ± SEM. Unpaired Student's $t$-test (two-tailed) was used to calculate $p$ values. *** and **** are for $p$ values ≤ 0.001 and 0.0001, respectively. IB immunoblot, KO knockout, Pro. protein, Inj. injection, KD knockdown.

NAADP-mediated Ca$^{2+}$ release[27]. Therefore, 100 nM NAADP in pipette solution, which upon microinjection induced a great (~fourfold) increase in the fluorescence of the Ca$^{2+}$ indicator (Fig. 2c, d and Supplementary Fig. 4a), was used for all the subsequent Ca$^{2+}$-imaging experiments of NAADP-evoked Ca$^{2+}$ release in this study. In agreement with the previously established characteristics of NAADP-evoked Ca$^{2+}$ release[6–9], the NAADP (100 nM in pipette solution)-elicited Ca$^{2+}$ signal was largely abolished in the absence of exogenous TPC expression or after cells had been pretreated with *trans*-Ned-19 (10 μM), or bafilomycin A1 (1 μM), an inhibitor of the endolysosomal H$^+$-ATPase that prevents pH-dependent Ca$^{2+}$ accumulation in endolysosomal stores (Supplementary Fig. 4a, c). Notably, the NAADP-elicited Ca$^{2+}$ increase was fully abolished in TPC2-

expressing Lsm12-KO cells as the Ca$^{2+}$ changes were similar to those observed without NAADP (vehicle only) in TPC2-expressing WT cells (Fig. 2c, d). The NAADP-evoked Ca$^{2+}$ release was restored in Lsm12-KO cells by co-transfection of TPC2 with Lsm12 (Fig. 2c, d). Similar results were obtained with TPC1-expressing cells in that Lsm12-KO abolished the NAADP-elicited Ca$^{2+}$ signal and reintroduction of Lsm12 expression largely restored the response (Fig. 2e and Supplementary Fig. 4c). In contrast, KO of Lsm12 expression had no influence on the intracellular Ca$^{2+}$ elevation induced by 1 μM extracellular ATP (Supplementary Fig. 4b), a purinergic receptor agonist that triggers IP$_3$ production and Ca$^{2+}$ mobilization from the ER. Thus, we conclude that Lsm12 is specifically required for NAADP-evoked Ca$^{2+}$ release.

To evaluate whether Lsm12 is directly involved in NAADP signaling, we obtained a purified recombinant human Lsm12 protein sample (hLsm12-His$_{E.coli}$) by expressing a 6×His-tagged human Lsm12 protein in *Escherichia coli* and purifying it with immobilized metal affinity chromatography. Notably, microinjection of the purified hLsm12-His$_{E.coli}$ protein (~4 μM in the pipette) together with NAADP immediately restored the NAADP-evoked Ca$^{2+}$ release in Lsm12-KO cells while the microinjection of the protein alone had no such rescuing effect (Fig. 2c, d and Supplementary Fig. 4c). Upon heat-inactivation treatment, the purified hLsm12-His$_{E.coli}$ protein became much less effective in restoration of the NAADP-evoked Ca$^{2+}$ release in Lsm12-KO cells (Fig. 2d and Supplementary Fig. 4c), indicating that the rescuing effect is directly related to the Lsm12 protein activity. These results showed that Lsm12 is essential and immediately participates in the process of NAADP-evoked Ca$^{2+}$ release in HEK293 cells, supporting a direct role of Lsm12 in NAADP signaling.

To examine whether Lsm12 is also important for NAADP signaling in other cell lines, we used siRNA to knockdown (KD) Lsm12 expression in SK-BR-3 cells, a breast cancer cell line that was previously shown to be responsive to NAADP[7]. We observed that Lsm12 expression at the protein level was greatly reduced in the Lsm12-KD cells (Fig. 2f). Correspondingly, the NAADP-evoked Ca$^{2+}$ elevation was also markedly decreased in the Lsm12-KD cells (Fig. 2g and Supplementary Fig. 4c). KD of Lsm12 expression in HEK293 cells also significantly reduced the NAADP-evoked Ca$^{2+}$ release elevation by 65% while KD of other two randomly selected Lsm proteins (Lsm5 and Lsm11) caused a much less (≤20%) reduction in the cells' response to NAADP (Supplementary Fig. 4d–f).

**Lsm12 directly binds to NAADP with high affinity.** To determine whether Lsm12 is truly an NAADP receptor, we performed competition ligand binding assays to examine its binding affinity to NAADP. A key feature of the unknown NAADP receptor in NAADP-evoked Ca$^{2+}$ release is its high and specific affinity to NAADP relative to the closely related NADP[6,22]. To exclude the possibility that some endogenous Lsm12-interacting protein of HEK293 cells might be involved in mediating the observed binding of NAADP to Lsm12, we performed binding assays using the purified hLsm12-His$_{E.coli}$ protein that was produced in a prokaryotic expression system and confirmed to be functional in NAADP signaling upon microinjection of the protein into HEK293 Lsm12-KO cells (Fig. 2b, c). We observed that hLsm12-His$_{E.coli}$ can be effectively pulled down by immobilized NAADP and that the formation of immobilized NAADP−Lsm12 complex was competitively prevented by the inclusion of low concentrations (nM range) of free NAADP (Fig. 3a and Supplementary Fig. 5a). The estimated $K_d$ for NAADP binding is ~30 nM, whereas NADP up to 100 μM had nearly no effect on the binding between hLsm12-His$_{E.coli}$ and immobilized NAADP (Fig. 3a and Supplementary Fig. 5a). Since $^{32}$P-NAADP-based radioligand binding was the common method used in determining the binding affinity of NAADP to endogenous receptors in HEK293 cells[6,22,28], we synthesized $^{32}$P-NAADP from $^{32}$P-NAD as previously described[28] and purified $^{32}$P-NAADP with thin-layer chromatography (Supplementary Fig. 5b). We observed that $^{32}$P-NAADP can effectively bind to hLsm12-His$_{E.coli}$ and that the binding can be competitively blocked by preincubation of the protein with nM concentrations of non-radiolabeled NAADP (Fig. 3b). The $^{32}$P-NAADP binding curve can be fitted with a high-affinity site ($K_d$ = ~20 nM; 79% in fraction) and a low-affinity site ($K_d$ = ~5 μM; 21% in fraction) (Fig. 3b). NADP up to 100 μM showed nearly no influence on the binding of $^{32}$P-

NAADP to hLsm12-His$_{E.coli}$ (Fig. 3b). We also performed NAADP binding assay with HEK293 cells by using immobilized NAADP to pull down endogenous NAADP binding proteins and free NAADP to compete in binding. The Lsm12 precipitated by immobilized NAADP was probed by an anti-Lsm12 antibody. Similarly, NAADP effectively reduced the Lsm12 protein precipitated by immobilized NAADP with an estimated $K_d$ of ~24 nM (~69% in fraction) (Fig. 3c and Supplementary Fig. 5c). However, NADP at much higher (~14×) concentrations ($K_d$ of ~340 nM; ~62% in fraction) can also compete to reduce the Lsm12 captured by immobilized NAADP (Fig. 3c and Supplementary Fig. 5c). It is unclear why NADP became more effective in its reduction of endogenous Lsm12 precipitation by immobilized NAADP as compared to that observed with purified hLsm12-His$_{E.coli}$. This could be due to some posttranslational modification or interacting protein of endogenous Lsm12 in HEK293 cells that enhances Lsm12's binding to NADP. A similar competitive effect of NADP was also previously described for the photoaffinity crosslinking of an NAADP analog to the 22/23-kDa doublet NAADP binding proteins in mammalian cells[21]. Overall, our results of competition ligand binding assays with purified and endogenous proteins demonstrate that Lsm12 is a high-affinity and selective receptor for NAADP.

**Lsm12 mediates NAADP signaling via its Lsm domain.** Lsm12 possesses an N-terminal Lsm domain and a putative C-terminal anticodon-binding (AD) domain[29] (Fig. 4a). To determine which domain or region of Lsm12 is involved in NAADP-evoked Ca$^{2+}$ release, we generated three truncation mutants ΔLsm, ΔAD, and Δlinker by deletion of the Lsm domain (residues 4–67), the C-terminus from the putative AD domain (residues 85–170) to the C-terminal end, and the linker region (residues 68–84) between the Lsm and AD domains, respectively. Upon transfection of the constructs into the Lsm12-KO cells, we observed that the ΔLsm mutant was unable to rescue the NAADP-evoked Ca$^{2+}$ release, whereas the ΔAD and Δlinker mutants remained largely functional in NAADP signaling (Fig. 4b and Supplementary Fig. 4c). To understand the function of the Lsm domain, we tested the mutants' function in binding to NAADP and TPC2. We found that the ΔLsm mutation abolished the association between $^{32}$P-NAADP and a recombinant FLAG-tagged Lsm12 protein that was heterologously expressed in Lsm12-KO cells and immunoprecipitated by an anti-FLAG antibody (Fig. 4c). Similarly, ΔLsm mutation eliminated the capability of the protein's binding to immobilized NAADP, whereas ΔAD and Δlinker mutations had no such effect (Fig. 4d). By co-immunoprecipitation, we also found that only ΔLsm caused a loss of the association between Lsm12 and TPC2 (Fig. 4e). Thus, the Lsm12's Lsm domain plays a dual role by interacting with both NAADP and TPC2. However, NAADP and TPC2 must interact with different regions of the Lsm domain since no drastic difference in the co-immunoprecipitation of Lsm12 and TPC2 was observed in the absence and presence of NAADP (Fig. 4f).

**Lsm12 mediates the apparent association of NAADP to TPC2 and TPC2-containing membranes.** The high binding affinity between NAADP and Lsm12 is comparable to the previously reported binding affinity between NAADP and TPC-enriching protein or membrane preparations[6,22]. We reasoned that the apparent association between NAADP and TPCs or TPC-containing cell membranes should result from Lsm12, which brings NAADP and TPCs together. To examine this possibility, we first determined the contribution of Lsm12 in NAADP binding to TPC2. We observed that both TPC1 and TPC2 were pulled down by immobilized NAADP but only in the presence of

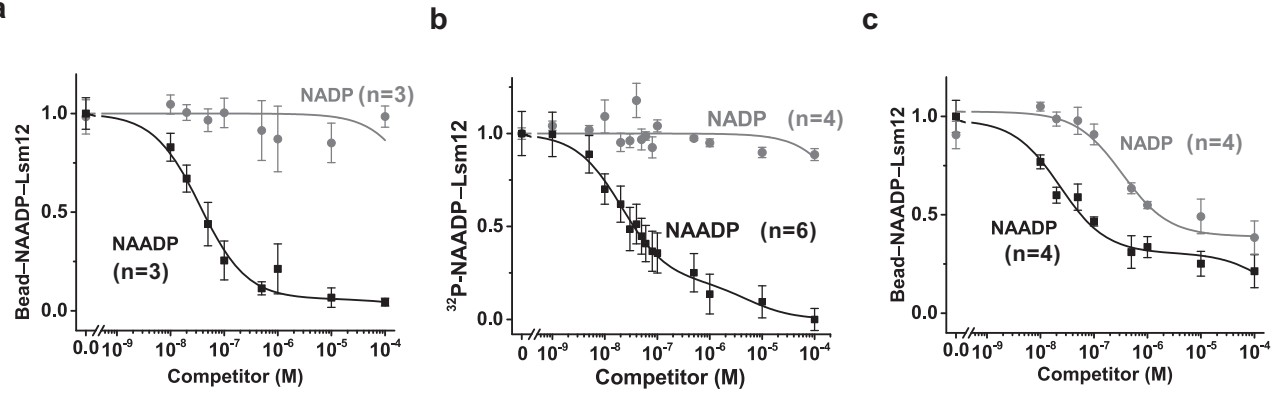

**Fig. 3 Lsm12 functions as a high-affinity NAADP receptor. a** Plot of the relationship between hLsm12-His$_{E.coli}$ pulled down by immobilized NAADP and the concentrations of free NAADP and NADP. **b** Competition radioligand binding assay of the association between $^{32}$P-NAADP and purified hLsm12-His$_{E.coli}$ in the absence or presence of various concentrations of non-radiolabeled NAADP and NADP. **c** Plot of the relationship between HEK293 endogenous Lsm12 pulled down by immobilized NAADP and the concentrations of free NAADP and NADP. Data were presented as mean value ± SEM.

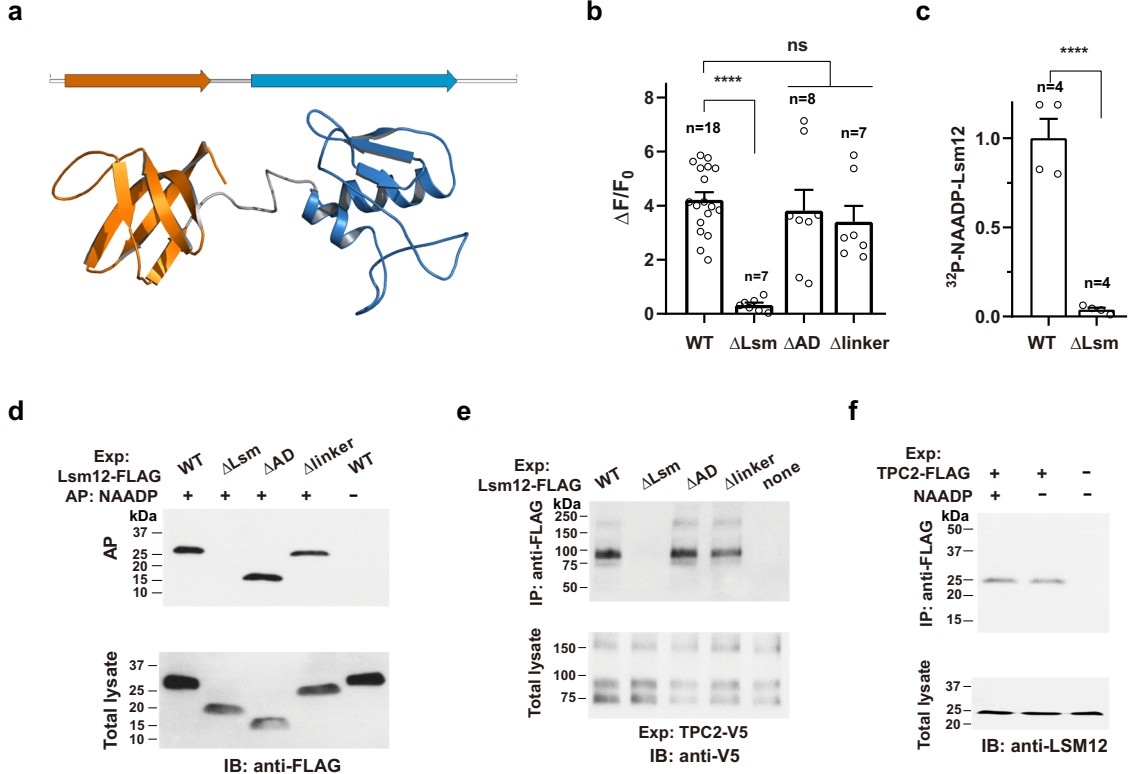

**Fig. 4 Lsm12 functions in NAADP signaling via its Lsm domain. a** Putative domain assignment and predicted 3D structural model for Lsm12 (GenBank: AAH44587.1). Region assignment: Lsm domain, residues 4–67; Linker, residues 68–84; putative AD domain, residues 85–170. The structure was drawn with a previously created model (https://swissmodel.expasy.org/repository/uniprot/Q3MHD2)[42]. **b** Averaged NAADP-induced changes in Ca$^{2+}$ indicator fluorescence in Lsm12-KO cells transfection with Lsm12 WT and mutant constructs. **c** Specific binding between $^{32}$P-NAADP and FLAG-tagged Lsm12 WT and ΔLsm proteins that were transiently expressed in Lsm12-KO cells and immunoprecipitated by an anti-FLAG antibody. **d** Immunoblot of Lsm12 WT and mutants pulled down by immobilized NAADP. **e** Co-immunoprecipitation between TPC2 and Lsm12 mutants. **f** Co-immunoprecipitation between TPC2 and Lsm12 in the absence or presence of 100 μM NAADP. Exp expression. Data were presented as mean value ± SEM. Unpaired Student's *t*-test (two-tailed) was used to calculate *p* values. **** is for *p* value ≤ 0.0001.

Lsm12 as immobilized NAADP pulled down neither TPC2 nor TPC1 from the Lsm12-KO cells (Fig. 5a). In contrast, immobilized NAADP pulled down Lsm12 in a largely TPC expression–independent manner since no drastic difference was observed in the absence or presence of exogenous TPC1 or TPC2 expression in HEK293 cells (Fig. 5b), which is also consistent with the above reference that NAADP and TPC2 interact with

different regions of the Lsm12's Lsm domain. We next determined the contribution of Lsm12 in NAADP binding to TPC2-expressing cell membranes using a competition radioligand binding assay as previously described[6,22]. We preincubated cell membranes of TPC2-expressing cells with different concentrations of non-radiolabeled NAADP before the addition of $^{32}$P-NAADP (~1 nM). The specific binding to the cell membranes was

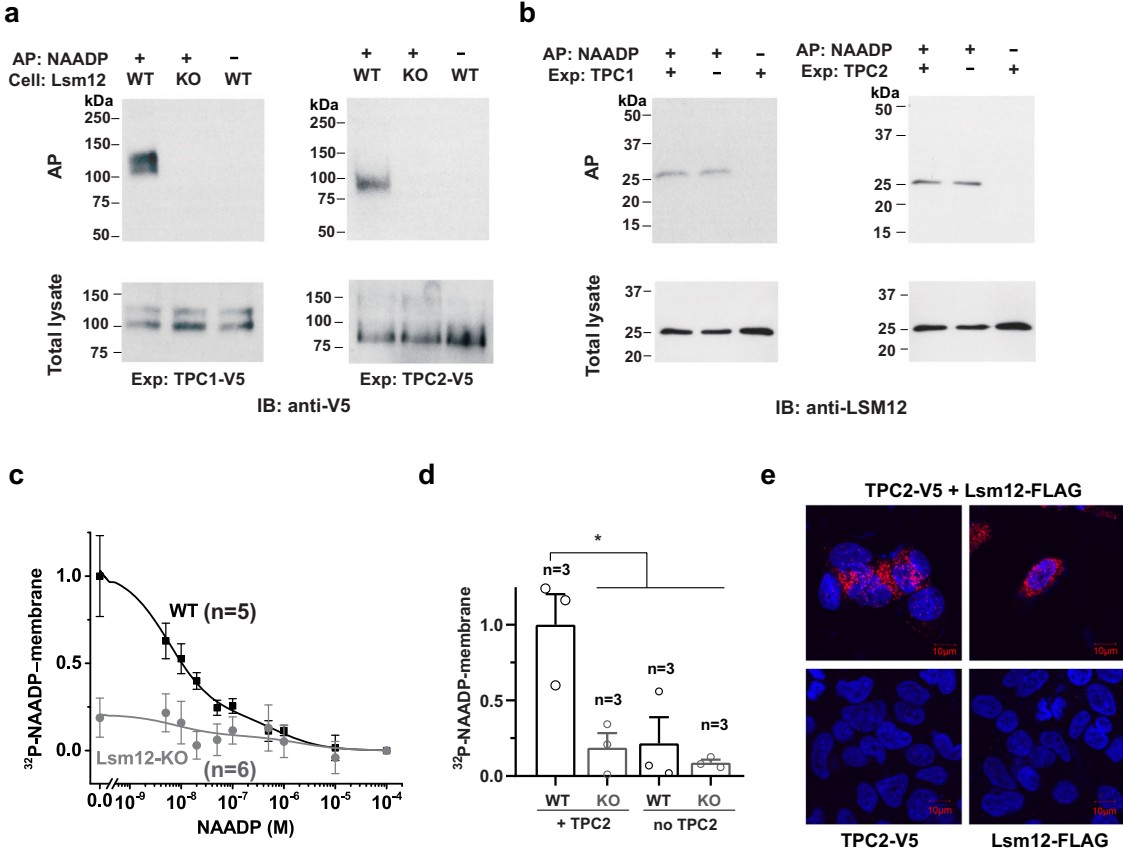

**Fig. 5 Lsm12 mediates the apparent association of NAADP to TPC2 and cell membranes. a** Immunoblot of TPC1 and TPC2 pulled down by immobilized NAADP from TPC1- and TPC2-expressing HEK293 WT and Lsm12-KO cells. **b** Immunoblot of Lsm12 pulled down by immobilized NAADP from cells with/without exogenous TPC1 and TPC2 expression. **c** Competition radioligand binding assay of the association between $^{32}$P-NAADP and TPC2-expressing cell membranes in the presence of various concentrations of non-radiolabeled NAADP. **d** Specific binding of $^{32}$P-NAADP to cell membranes in WT and Lsm12-KO cells with/without exogenous TPC2 expression. **e** In situ PLA of TPC2 and Lsm12 colocalization in HEK293 cells. Top panels show cells expressing both exogenous TPC2 and Lsm12 and bottom panels show cells expressing only one of them. AP affinity precipitation, Exp expression, AP: NAADP± AP with/without immobilized NAADP. Data were presented as mean value ± SEM. Unpaired Student's *t*-test (two-tailed) was used to calculate *p* values. * is for *p* values ≤ 0.05.

estimated by subtracting the bound total $^{32}$P from that of non-specifically bound $^{32}$P determined in the presence of 100 μM non-radiolabeled NAADP. Similar to the previous report[6], we observed a high-affinity $^{32}$P-NAADP binding of TPC2-expressing membranes that was competed off by 40−75% with the pre-incubation of 5−50 nM non-radiolabeled NAADP (Fig. 5c). However, in membranes prepared from TPC2-expressing Lsm12-KO cells, the specific binding of $^{32}$P-NAADP was mostly lost (e.g., by ~80% in the absence of non-radiolabeled NAADP) (Fig. 5c, d). Given that Lsm12 is a cytosolic protein (Fig. 2a), an Lsm12-interacting protein on the membrane is expected to be needed for the apparent association of NAADP to cell membranes. We found that TPC2 expression was also required for the association of $^{32}$P-NAADP to cell membranes (Fig. 5d), suggesting that Lsm12 is mostly associated with the membrane through its interaction with TPCs. These results demonstrate that TPCs are not NAADP receptors and that the apparent association of NAADP to TPCs or TPC-containing membranes was indeed mediated by Lsm12.

To evaluate whether Lsm12 and TPCs can colocalize inside cells, we performed in situ proximity ligation assay (PLA), which is an established method of visualizing protein–protein interactions in cells[30]. This assay combines antibody-based protein recognition and nucleotide-based rolling circle amplification to allow a protein complex to be fluorescently labeled only when the

two epitopes on the interacting proteins are in close proximity. Upon co-expression of the V5-tagged TPC2 and FLAG-tagged Lsm12 in HEK293 cells, in situ PLA with anti-V5 and anti-FLAG antibodies under a cell permeabilized condition produced punctuated PLA signals inside cells, agreeing with the intracellular lysosomal localization of TPC2 (Fig. 5e). Such PLA signals should be originated from TPC2-Lsm12 complex formation as they were absent if only one of the constructs was expressed (Fig. 5e), suggesting that TPC2 and Lsm12 can closely colocalize in cells.

**Lsm12 is required for NAADP-induced TPC activation**. TPCs have been considered as the key lysosomal cation channels responsive to NAADP stimulation for intracellular Ca$^{2+}$ release. However, controversy exists as the TPCs' sensitivity to NAADP was not observed in multiple reports[17–20]. Similarly, we were also unable to detect any NAADP-activated TPC2 currents on excised plasma membrane patches in an inside-out recording of the plasma membrane-targeted mutant TPC2$^{PM}$ (TPC2$^{L11A/L12A}$)[8] channels. To retain the NAADP sensitivity of the channels, we measured NAADP microinjection-induced whole-cell currents of voltage-clamped HEK293 cells that expressed TPC2$^{PM}$ (Fig. 6a). With this new method combining whole-cell current recording with NAADP microinjection, we observed robust NAADP-

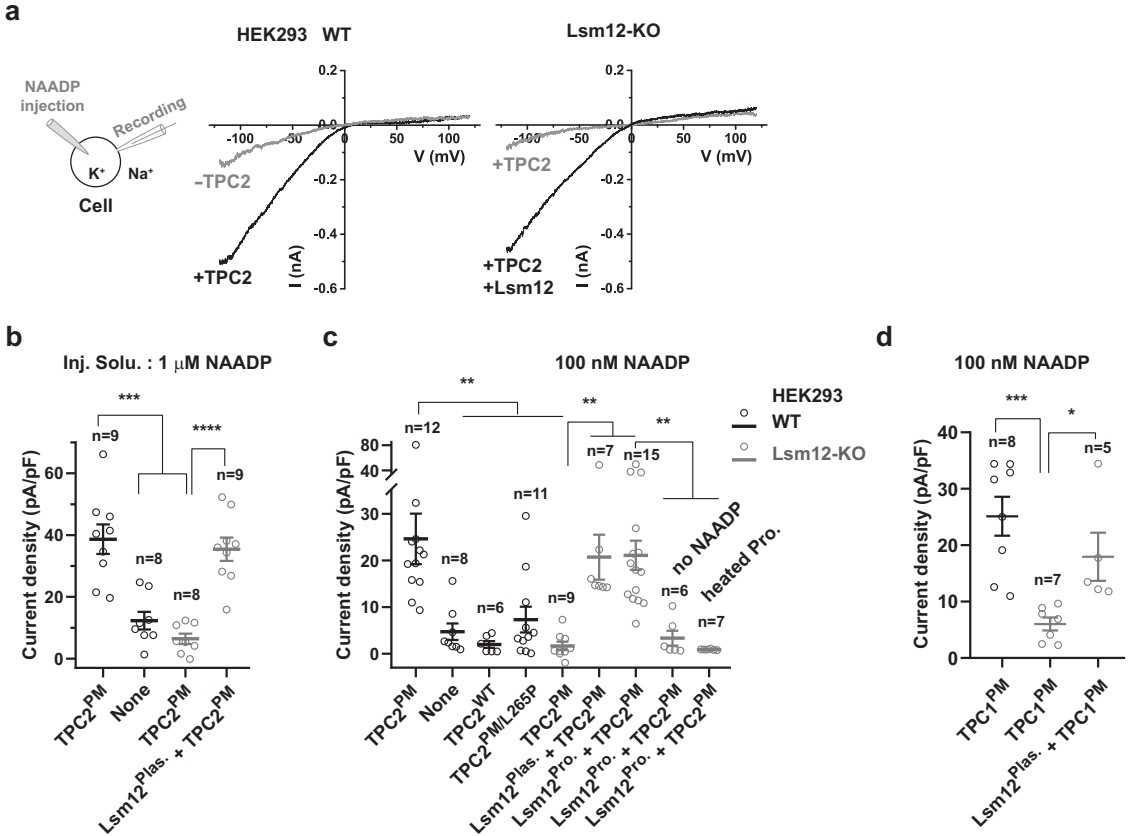

**Fig. 6 Lsm12 is required for TPC activation by NAADP. a** Averaged traces of NAADP (1 μM in pipette solution) microinjection-induced whole-cell currents (averaged) in HEK293 WT and Lsm12-KO cells transfected with/without TPC2[L11A/L12A] and Lsm12. **b–d** Averaged current density of NAADP microinjection-induced whole-cell TPC2 (**b**, **c**) and TPC1 (**d**) currents in HEK293 WT and Lsm12-KO cells. The cells (WT shown in black and Lsm12-KO in gray) were transfected with/without Lsm12 plasmid (Lsm12[Plas.]) and plasmid of TPC2 WT (TPC2[WT]) or plasma membrane-targeted mutant (TPC2[PM] or TPC1[PM]). NAADP was injected at 1 μM (**b**) or 100 nM (**c**, **d**). The purified Lsm12 protein (Lsm12[Pro.] in **c**) was injected at ~100 ng/μl (~4 μM) in injection pipette solutions. Data were presented as mean value ± SEM. Unpaired Student's t-test (two-tailed) was used to calculate p values. *, **, ***, and **** are for p values ≤ 0.05, 0.01, 0.001, and 0.0001, respectively. Inj. Solu. injection solution, Pro protein.

evoked Na$^+$ currents (inward) which were ~0.5 nA or 38.7 ± 4.8 pA/pF on average at −120 mV for an injection solution containing 1 μM NAADP (Fig. 6a, b). The recorded currents were slightly smaller (~0.3 nA or 24.6 ± 5.4 pA/pF at −120 mV) when the injection solution contained 100 nM NAADP (Fig. 6b), a concentration that can achieve maximal NAADP-evoked Ca$^{2+}$ release in our Ca$^{2+}$-imaging experiments (Fig. S4a). Such currents were mostly absent or reduced when no exogenous TPC2 was expressed (Fig. 6a–c), when the lysosome-targeted WT TPC2 was used (Fig. 6c), or in the presence of a dominant-negative mutation L265P[8] (Fig. 6c), indicating that the NAADP-evoked inward Na$^+$ currents were mostly originated from the plasma membrane-targeted TPC2[PM] channels. Consistent with Lsm12 as the NAADP receptor, the NAADP-induced TPC currents in TPC2-expressing Lsm12-KO cells were absent (Fig. 6a–c). The response was restored by supplementing Lsm12 via transfection of an Lsm12-expressing plasmid (Fig. 6a–c) or co-microinjection of NAADP with the purified Lsm12 protein (Fig. 6c). Micro-injection of the purified Lsm12 protein alone without NAADP or heat-inactivated Lsm12 protein together with NAADP didn't result in a significant increase in the inward Na$^+$ currents (Fig. 6c), suggesting that an active Lsm12 protein is immediately required for TPC2 activation by NAADP. We also performed a whole-cell patch-clamp recording of the plasma membrane-targeted TPC1[PM] (TPC1[L11A/I12A])[31] channel. We similarly observed that the TPC1[PM] channel can be activated by micro-injected NAADP in WT cells but not in Lsm12-KO cells and that

reexpression of Lsm12 in the KO cells can largely restore the NAADP-induced TPC1[PM] activation (Fig. 6d). Given that PI(3,5)P$_2$ directly binds to TPCs, it is anticipated that Lsm12 is not required for TPC activation by PI(3,5)P$_2$. Consistently, we observed that the dose responses of TPC2 to PI(3,5)P$_2$ were not significantly different in the presence and absence of Lsm12 expression when the channel currents were recorded by whole lysosome (enlarged) patch-clamp recording (Supplementary Fig. 6a). Thus, as is expected for the function of Lsm12 as an NAADP receptor and an interacting protein of TPCs, these results demonstrate that Lsm12 mediates TPC2 and TPC1 activation by NAADP.

**Lsm12 is important for NAADP signaling in mouse embryonic fibroblasts.** To evaluate the importance of Lsm12 in primary cells, we generated Lsm12 mutant mice with the CRISPR/Cas9 method. The ablation of Lsm12 function by a frameshifting mutation, a deletion of 18 bp (ATGTCCCTCTTCCAGTGG) in the third exon, appeared to be embryonically lethal as no homozygous mouse could be obtained. Thus, a mutant mouse harboring a different 18 bp (TCTTCCAGTGGAAACCC) dele-tion in the third exon was used. This non-frameshifting mutation, designated as Lsm12[Δ45–50], results in a deletion of six residues (SSSGKP[50]) in the Lsm domain. Given that mouse embryonic fibroblasts (MEFs) were frequently used in studying NAADP signaling or TPC function in primary cells[11,32–34], we isolated

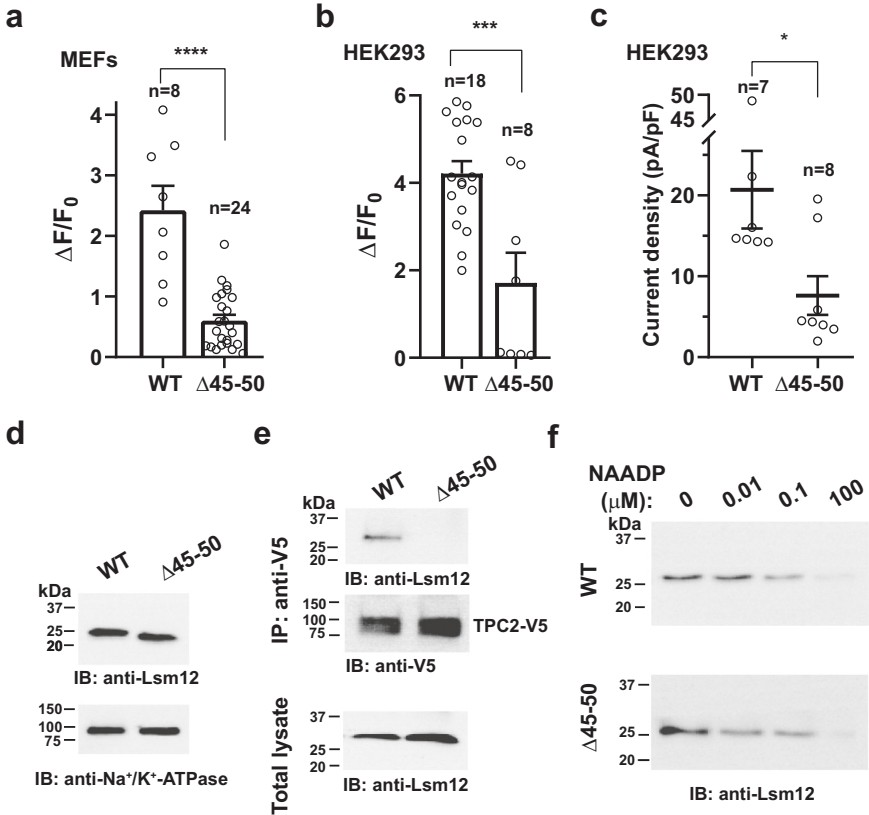

**Fig. 7 Lsm12 function in NAADP signaling is compromised by Δ45–50 mutation in MEFs and HEK293 cells. a** Averaged NAADP-induced changes in $Ca^{2+}$ indicator fluorescence in MEFs prepared from WT and $Lsm12^{\Delta45-50}$ mutant mice. **b** Averaged NAADP-induced changes in $Ca^{2+}$ indicator fluorescence in HEK293 Lsm12-KO cells transiently expressing TPC2 and Lsm12 WT or Δ45–50 mutant. **c** Averaged current density of NAADP (100 nM in pipette solution) microinjection-induced whole-cell currents in HEK293 Lsm12-KO cells transiently expressing $TPC2^{PM}$ and Lsm12 WT or Δ45–50 mutant. **d** Immunoblot of Lsm12 in MEFs prepared from WT and $Lsm12^{\Delta45-50}$ mutant mice. MEFs were prepared and pooled from five WT and six $Lsm12^{\Delta45-50}$ mutant mice, respectively. **e** Co-immunoprecipitation (IP) of TPC2 (V5-tagged) and LSM12 WT or Δ45–50 mutant transiently expressed in HEK293 LSM12-KO cells. **f** Immunoblot of Lsm12 WT and Δ45–50 mutant that was transiently expressed in HEK293 Lsm12-KO cells and pulled down by immobilized NAADP in the presence of different concentrations of free NAADP. Data were presented as mean value ± SEM. Unpaired Student's *t*-test (two-tailed) was used to calculate *p* values. **\***, **\*\*\***, and **\*\*\*\*\*** are for *p* values ≤ 0.05, 0.001, and 0.00001, respectively. IB immunoblot, IP immunoprecipitation.

MEFs from WT and $Lsm12^{\Delta45-50}$ mice and measured the cells' response to NAADP. We observed that the NAADP-evoked $Ca^{2+}$ elevation was greatly reduced by 75% in MEFs of the $Lsm12^{\Delta45-50}$ mice as compared to that of WT mice (Fig. 7a and Supplementary Fig. 4c). When the $Lsm12^{\Delta45-50}$ mutant construct was transiently expressed in the HEK293 Lsm12-KO cells, it was also much less effective (a reduction by ~60%) than WT Lsm12 in rescuing the NAADP-evoked $Ca^{2+}$ release in the HEK293 Lsm12-KO cells (Fig. 7b and Supplementary Fig. 4c). Consistently, the NAADP-induced TPC2 currents were not effectively rescued by the $Lsm12^{\Delta45-50}$ mutant (~60% reduction in efficiency as compared to WT) upon its transient expression in HEK293 Lsm12-KO cells (Fig. 7c). The deficiency of the $Lsm12^{\Delta45-50}$ mutant in supporting NAADP-induced $Ca^{2+}$ release and TPC2 current activation was not caused by reduced expression as the mutant and WT were similarly expressed in MEFs (Fig. 7d) and the mutant displayed a higher transient expression level than WT in HEK293 Lsm12-KO cells (Fig. 7e). The latter could result in an underestimate of the mutant's deficiency in HEK293 cells. These results obtained with MEFs and HEK293 cells show deficiency of the $Lsm12^{\Delta45-50}$ mutant in NAADP signaling and support a key role of Lsm12 in NAADP signaling in MEFs.

To understand the mechanisms underlying the $Lsm12^{\Delta45-50}$ mutant's defects in NAADP-induced $Ca^{2+}$ release and TPC2

activation, we assayed its functions in TPC and NAADP binding. We found that its association with TPC2 is compromised. Co-immunoprecipitation of the co-expressed TPC2 and Lsm12 showed that the Δ45–50 mutant was barely pulled down together with TPC2 (Fig. 7e). The $Lsm12^{\Delta45-50}$ mutant remained functional in its binding with NAADP as 10 and 100 nM NAADP still competitively reduced the protein's association with immobilized NAADP in a manner similar to that observed with the WT protein (Fig. 7f). These suggest that the complex formation between Lsm12 and TPC2 is likely required in NAADP-induced TPC2 activation and $Ca^{2+}$ mobilization from intracellular stores.

## Discussion

The unresolved molecular identity of the NAADP receptor has limited our understanding of the molecular basis and mechanisms of NAADP-evoked $Ca^{2+}$ release and signaling. The present study has identified a previously largely uncharacterized protein, Lsm12, as an NAADP receptor essential for NAADP-evoked $Ca^{2+}$ release. According to The Human Protein Atlas (www.proteinatlas.org), Lsm12 at the RNA level is ubiquitous and abundantly expressed in different tissues and cell lines. Given that Lsm12 is functionally critical for NAADP-evoked $Ca^{2+}$ release in all three tested cell types (HEK293, SK-BR-3, and MEFs), we

anticipate that Lsm12 as an NAADP receptor plays a critical role in NAADP signaling in different cells.

Since the reports of a key role of TPCs in NAADP signaling[6,7], there has been some ambiguity whether TPCs are NAADP receptors[21,24] and whether NAADP can activate TPCs[17–20,24]. Our identification of Lsm12 as an NAADP receptor and characterization of its relationship to TPCs have clearly consolidated the notion that TPCs are not direct receptors of NAADP[21,23,24]. We were also able to observe Lsm12-dependent NAADP-induced TPC2 and TPC1 currents in whole-cell patch-clamp recording by intracellular application of NAADP via microinjection, a method same as we used in the $Ca^{2+}$-imaging analysis of NAADP-evoked $Ca^{2+}$ release. This NAADP-microinjection method allows us to obtain the baseline currents in the absence of drug application and a click-speed drug delivery directly inside cells, which could be particularly helpful if the response to a drug is small or transient that might be hard to observe when the drug is applied by the common dialysis method via recording pipette. Furthermore, the physical association between Lsm12 and TPC2 likely plays a role in NAADP signaling as the Lsm12$^{\Delta45-50}$ mutant, which has a weakened association with TPC2, was deficient in NAADP-evoked TPC2 activation and intracellular $Ca^{2+}$ elevation. These led us to conclude that TPCs are regulated by NAADP via the receptor Lsm12 (Supplementary Fig. 6b).

Lsm12 belongs to the Lsm protein family, which includes smaller Lsm proteins (Lsm1–10) that contain only an Lsm domain and the larger Lsm proteins (Lsm11, 12, 14A, 14B, and 16 in humans) that contain an extra C-terminal non-Lsm domain[29]. Although much less or little is known about the functions of larger Lsm proteins, smaller Lsm proteins (e.g., Lsm1–8) generally function as scaffolds or chaperones to bind to RNA oligonucleotides, facilitating RNA assembly, modification, storage, transportation, or degradation and thus globally affecting cell gene expression[35]. We have found that the function of Lsm12 in NAADP signaling is mainly mediated by its Lsm domain in both NAADP binding and association with TPC2. The Lsm domain is predicted to form a highly conserved tertiary structure of the Lsm fold that consists of a five-stranded antiparallel β-sheet (Fig. 4a). As seen in many other Lsm proteins, the Lsm domain may oligomerize into an Lsm ring that allows binding to an oligonucleotide (adenine or uracil)[35]. Given that NAADP is essentially a type of dinucleotide, Lsm12 is well-positioned to serve as an NAADP receptor via its Lsm domain. It remains unknown whether some other Lsm proteins could also function as an NAADP receptor. Although other Lsm proteins are also expressed in HEK293 cells, Lsm12 likely functions as a dominant NAADP receptor because (1) Lsm12-KO fully eliminates the NAADP response; (2) no other Lsm protein has been identified as NAADP and TPC-interacting proteins in our proteomic analyses; (3) and KD of the other two tested Lsm proteins, the smaller Lsm5, and the larger Lsm11 had greatly smaller effects on the cells' NAADP response.

In spite of its potential RNA-binding function which might have a global impact on protein expression, evidence from this study strongly supports that Lsm12 directly participates in NAADP-evoked $Ca^{2+}$ release. These include: (1) Lsm12 is a selective and direct high-affinity receptor for NAADP; (2) Lsm12 mediates the apparent association of NAADP to TPCs and the NAADP-induced activation of TPCs; (3) Importantly, microinjection of the purified Lsm12 protein immediately restored the NAADP-evoked $Ca^{2+}$ release and TPC2 activation in Lsm12-KO cells, suggesting that the loss of NAADP response in the Lsm12-KO cells is unlikely caused by some defect in cellular structure or content whose restoration process is usually not immediate, e.g., ~5 min for lysosomal $Ca^{2+}$ refilling if depleted[36].

A distinct protein, JPT2, was recently identified as an NAADP binding protein in erythrocytes[37] and human Jurkat T cells[38], following up the previous report of the 22/23-kDa doublet proteins that were photocrosslinked to 5-azido-NAADP[21]. JPT2 seems to be functionally different from Lsm12 in NAADP signaling. Of note, JPT2 regulated NAADP-induced $Ca^{2+}$ microdomain formation in a RyR1—but not TPC-dependent manner in human Jurkat and primary rat T cells[38]. JPT2 co-immunoprecipitated with TPC1 but not TPC2[37]. KD of JPT2 partially reduced the photocrosslinked 23 kDa proteins[37], implying the presence of other NAADP binding proteins at a similar protein size. Although it remains unclear whether JPT2 can mediate NAADP-evoked TPC activation and/or TPC-dependent $Ca^{2+}$ release, the identification of JPT2 in other cell types with a different NAADP analog supports the presence of different NAADP binding proteins in different cells or different NAADP signaling pathways.

Overall, our studies reveal the molecular identity of an NAADP receptor to be an Lsm protein (Lsm12) and establish its essential role in NAADP-evoked TPC activation and $Ca^{2+}$ mobilization from intracellular stores. Our findings thus provide a new molecular basis toward elucidating the mechanisms and function of NAADP signaling. Future studies will be needed to elucidate the detailed molecular mechanisms underlying the TPC activation caused by the NAADP binding to Lsm12. Since many other Lsm proteins are also well expressed at the RNA level in different cells, more studies will be needed to determine whether some other Lsm proteins can also function in NAADP signaling. Given that Lsm12 is also a putative RNA binding and regulation protein, an investigation of the potential crosstalk between NAADP signaling and RNA regulation will be necessary.

## Methods

**Mammalian cell lines, culture, and molecular biology**. A variant of the HEK293 cell line, 293H (Invitrogen), which has better adherence in monolayer culture, was used throughout the study. SK-BR-3 cells were obtained from ATCC. HEK293 and SK-BR-3 cells were cultured in Dulbecco's modified Eagle's medium and RPMI 1640 medium, respectively, supplemented with 10% fetal bovine serum, 50 units/ml penicillin, and 50 mg/ml streptomycin. For quantitative proteomic analysis of NAADP and TPC-interacting proteins, HEK293 cells were cultured in SILAC-compatible DMEM supplemented with 10% dialyzed fetal bovine serum and 100 mg/L $^{13}C$ or $^{12}C$-labeled lysine and arginine (all reagents were from Thermo Fischer Scientific) for at least 6 days to achieve an incorporation efficiency of >90%.

Recombinant cDNA constructs of human TPC1 (GenBank: AY083666.1) and TPC2 (GenBank: BC063008.1) with/without FLAG and/or V5 epitopes on their C-termini were constructed with pCDNA6 vector (Invitrogen). The TPC1 and TPC2 constructs with an additional C-terminal tag of eGFP were used in the proteomic experiments. To facilitate the identification of transfected cells, an IRES-containing bicistronic vector, pCDNA6-TPC2-V5-IRES-AcGFP, was also generated and used in electrophysiological experiments. Recombinant human Lsm12 (GenBank: AAH44587.1) -expression plasmids were obtained either commercially (pCMV-Lsm12-Myc-FLAG from OriGene) or constructed with pCDNA6 (pCDNA6-Lsm12-FLAG-His). The plasmid (pGP-CMV-GCaMP6f) expressing the $Ca^{2+}$ reporter GCaMP6f[26] was obtained from Addgene (Cat # 40755). Cells were transiently transfected with plasmids with transfection reagent of Lipofectamine 2000 (Invitrogen), PolyFect (QIAGEN), or FuGENE HD (Promega) and subjected to experiments within 16–48 h after transfection.

An Lsm12-KO cell line of HEK293 cells was generated with Synthego's chemically modified sgRNA 5′-CCAGAAUGUCCCUCUUCCAG-3′ and GeneArt Platinum Cas9 nuclease (ThermoFisher Scientific) that were transfected together into cells using Lipofectamine CRISPRMAX Cas9 transfection reagent (ThermoFisher Scientific). KD of Lsm12, Lsm5, and Lsm11 expression in SK-BR-3 and HEK293 cells was achieved with predesigned dicer-substrate siRNA (DsiRNA) (Cat # hs.Ei.LSM12.13.2, hs.Ri.LSM5.13.2, and hs.Ri.LSM11.13.2 from IDT). A non-targeting DsiRNA (Cat # 51-01-14-03 from IDT) was used as a negative control. The real-time PCR assay was conducted in a MiniOpticon system (Bio-Rad Inc.) using KiCqStart (Millipore Sigma) SYBR Green ReadyMix (Cat # KCQS00) and predesigned primer pairs: TCGTGATGAAGAGTGATAAGG and CAAACTCAGTGACATCTTCC for Lsm5; AAAGCATATGAACGGGATTC and AGTGGATGTCTGTGAGTATC for Lsm11; and GCTTTAAAATGTCCCTCTT CC and GGCAAGCTTACTAACATTGAG for Lsm12. The real-time PCR reactions were performed as 95 °C for 30 s and then 34 cycles of 95 °C for 15 s, 58 °C for 30 s, and 72 °C for 30 s. The specificity and efficiency of the primers were

analyzed by melting curve analysis and reactions with serially diluted template DNA, respectively.

### Quantitative proteomic analyses of NAADP- or TPC-interacting proteins.

Immobilized NAADP was used for affinity-precipitation of NAADP-interacting proteins from HEK293 cells transfected with constructs of TPC1-eGFP-FLAG or TPC2-eGFP-FLAG. Immobilized NAADP was prepared as previously reported[25]. Briefly, 1 μmol of NAADP was incubated with 100 μl of 100 mM sodium periodate/100 mM sodium acetate (pH 5.0) mixture in the dark for 1 h. The unreacted sodium periodate was removed by adding 200 mM potassium chloride to the solution, incubating on ice for 5 min, and then centrifuging at 170,000x$g$ for 10 min. The supernatant was then incubated with sodium acetate, which was pre-balanced with adipic acid dihydrazide-Agarose beads (Cat# A0802 from Sigma-Aldrich) at 4 °C in the dark for 3 h. The NAADP-conjugated beads were then washed sequentially with 1 M NaCl and phosphate-buffered saline (PBS). Anti-FLAG antibody was immobilized to protein-A agarose beads without chemical crosslinking.

For affinity-precipitation, proteins (~5 mg) were solubilized from five dishes (round, 10 cm in diameter) of the heavy- and light-labeled cells, respectively, with a lysis buffer containing 2% dodecyl maltoside in 150 mM NaCl and 20 mM Tris-HCl (pH 7.4). The solubilized proteins were collected as supernatants after centrifugation at 17,000x$g$ for 10 min, and then incubated overnight with beads of immobilized NAADP or anti-FLAG antibody at 4 °C. A protease inhibitor cocktail (cOmplete, EDTA free; Cat # 11697498001 from Roche) was added in all buffers throughout the affinity-precipitation according to the manufacture's instruction. After being washed with lysis buffer three times, the bound proteins were eluted with 100 μg/ml FLAG peptide for collecting interacting proteins of FLAG-tagged TPCs, or NAADP (100 μM), and then 4% SDS for collecting NAADP-interacting proteins.

For SILAC quantification by mass spectrometry, the heavy- and light-labeled eluates originated from the same amount of total protein inputs (supernatants of cell lyses) were merged. After a brief separation by a short run of SDS-PAGE, the protein-containing gels were excised into three or five different bands. Each gel band was thoroughly washed with 50% acetonitrile in 25 mM ammonium bicarbonate and then subjected to in-gel reduction with dithiothreitol, alkylation with iodoacetamide, and digestion with trypsin (Mass Spectrometry Grade; Cat # V5280 from Promega) as we previously described[39]. Digested peptide mixtures were extracted, dried in a speed vacuum concentrator, reconstituted in 2% acetonitrile and 0.1% trifluoroacetic acid, and subjected to LC-MS/MS analysis with either a Q-Exactive Orbitrap mass spectrometer or an Orbitrap Fusion Lumos mass spectrometer (ThermoFisher Scientific). The mass spectrometer was operated in positive ion mode. MS/MS spectra were acquired with either higher-energy collisional dissociation (HCD) in Orbitrap (Q-Exactive) or collision-induced dissociation (CID) in ion trap (Fusion Lumos). The spectra labeling, database search, and quantification were carried out with Mascot Distiller (v 2.6) and Mascot (v 2.4) (MatrixScience) by searching against SwissProt (2016_04) human protein database (20,200 sequences). Database searches were performed with a peptide mass tolerance of 10 or 20 ppm and MS/MS tolerance of 20 mmu (Q-Exactive) or 0.5 Da (Fusion Lumos), allowing two missed cleavage sites and variable modifications of carbamidomethyl (C), oxidation (M), and N-terminal pyroglutamate (pyro-Glu). Quantification was obtained with Mascot Distiller with quantitation method of SILAC (K + 6, R + 6) from spectra pairs whose correlation coefficient between the observed precursor isotope distributions and the predicted ones is ≥0.6 and whose standard error after the fit to the signal intensities of each precursor pair is ≤1. For peptide sequence match (PSM), an identity threshold ($E$ value ≤0.05) was required for all assigned spectra. Only proteins with heavy:light ratio of ≥3 and a mascot score of ≥40 were reported as differential or putative interacting proteins. The false discovery rate (FDR) was estimated to be <1% for peptides and <3% for proteins. The TPC (TPC1 and TPC2) and NAADP interactomes were identified from protein samples of five replications and single preparation, respectively.

### Immunoprecipitation, immunoblotting, and immunofluorescence analyses.

Immunoprecipitation, immunoblot, and immunofluorescence analyses of protein interactions and expression were performed similarly to analyses we reported[39]. Rabbit polyclonal anti-FLAG antibody (Cat# F7425 from Sigma-Aldrich) and mouse anti-V5 (clone V5-10) agarose affinity gel (Cat# A7345 from Millipore) were used in immunoprecipitation. FLAG- or V5-peptide was used to elute the antibody-trapped proteins from the beads. Mouse monoclonal anti-FLAG M2 antibody (Cat# F3165 from Sigma-Aldrich) at 1:1000 dilution and mouse monoclonal anti-V5 antibody (Cat# R96125 from Invitrogen) at 1:10,000 dilution were used for immunoblotting. Rabbit monoclonal anti-LSM12 antibody (Cat# EPR12282 from Abcam) at a dilution factor of 1:1000 and 1:100 was used for immunoblotting and immunofluorescence, respectively.

### In situ PLA.

In situ PLA was performed using Duolink In Situ PLA reagents (Sigma-Aldrich) as we reported[39]. HEK293 cells transfected with FLAG-tagged Lsm12 and/or V5-Tagged TPC2 constructs were fixed in 4% paraformaldehyde 16–24 h after transfection. After permeabilization treatment (0.05% Triton X-100), cells were

incubated with mouse monoclonal anti-V5 (1:200 dilution; Cat# SAB2702199 from Sigma-Aldrich) and rabbit anti-FLAG (1:500 dilution; Cat# F7425 from Sigma-Aldrich) in phosphate-buffed saline (PBS) at room temperature for 1 h. HEK293 cells probed with primary antibodies were first incubated with secondary anti-mouse and anti-rabbit antibodies conjugated with oligonucleotides of a PLA probe and then subjected to oligonucleotide hybridization, ligation, amplification, and detection following the manufacturer's instructions. Finally, cells were mounted on slides using a mounting medium with DAPI and observed under a confocal microscope (FV1000; Olympus).

### Expression and purification of recombinant human Lsm12 with *E. coli*.

The bacterial expression plasmid pET-6His-TB-Lsm12, which encodes human Lsm12 protein with an N-terminally tagged cleavable thrombin (TB) cleavage motif after 6×His tag, was constructed with bacterial pET vector and transformed into *E. coli* strain BL21 (DE3). The cells were grown at 37 °C and collected 4 h after induction with 1 mM IPTG. Cells were harvested by centrifugation and broken by sonication in 50 mM Tris-HCl pH 8.0, 500 mM NaCl, and 5% glycerol supplemented with 1 mM PMSF and Pierce™ Protease inhibitor tablet (Cat # A32965 from Thermo-Fisher). After centrifugation at 16,000x$g$ for 40 min, the soluble fraction was collected as supernatant. The 6×His-tagged LSM12 (hLsm12-His$_{E.coli}$) was purified by passing the soluble fraction through a Ni-NTA column and then eluting the column with 300 mM imidazole in the same buffer. Imidazole and excessive salts in elute were removed by dialysis in 25 mM Tris-HCl pH 8.0 and 50 mM NaCl.

### NAADP binding assays.

The NAADP binding assay with immobilized NAADP was performed in a similar manner as that described above for affinity-precipitation of NAADP-interacting proteins except that the NAADP-conjugated beads were preincubated with various concentrations of label-free/unmodified NAADP or NADP before incubation with purified hLsm12-His$_{E.coli}$ (5 nM) or HEK293 cell lysis. The $^{32}$P-NAADP was synthesized from $^{32}$P-NAD (Cat# BLU023 from PerkinElmer Co.) following two sequential enzymatic reactions[28]. $^{32}$P-NAD was first catalyzed into $^{32}$P-NADP with 0.5 U/ml NAD kinase (Cat# AG-40T-0091from Adipogen Co.) in the presence of 5 mM Mg$^{2+}$-ATP and 100 mM HEPES, pH 7.4 at 37 °C for 4 h. The synthesized $^{32}$P-NADP was converted into $^{32}$P-NAADP with 1 μg/ml ADP-ribosyl cyclase (Cat# A9106 from Sigma-Aldrich) and 100 mM nicotinic acid. The synthesized $^{32}$P-NAADP was validated and purified with polyethyleneimine cellulose thin-layer chromatography developed in a solvent system of isobutyric acid−500 mM NH$_4$OH (5:3 v/v). For the assay involving $^{32}$P-NAADP binding to purified hLsm12-His$_{E.coli}$, the protein at 30 nM was incubated with various concentrations of non-radiolabeled NAADP or NADP for 30 min and then further incubated with 1 nM $^{32}$P-NAADP at room temperature for 1 h. The NAADP-protein complex was captured by Ni-NTA beads, washed three times with TBS, and eluted with 300 mM imidazole in TBS. For the assay involving $^{32}$P-NAADP binding to cell membranes, we followed a previously reported procedure[6] with slight modifications. Cell membranes corresponding to 1 mg protein were incubated with 1 nM $^{32}$P-NAADP after preincubation with non-radiolabeled NAADP/NADP. Unbound $^{32}$P-NAADP was separated from the membrane through rapid vacuum filtration through GF/B filter paper. The radioactivity in the eluates or filter papers was counted with a liquid scintillation counter.

### Generation of Lsm12$^{Δ45-50}$ mutant mice and preparation of MEFs.

The founder lines of Lsm12 mutant mice were generated using the CRISPR/Cas9 method via pronuclear injection of a mixture of sgRNA 5′-CCAGAAUGUCCCUCUUCCAG-3′ (Synthego, unmodified) and Cas9 into embryonic stem cells derived from C57BL/6 mice at the MD Anderson Cancer Center Genetically Engineered Mouse Facility. Homozygous Lsm12$^{Δ45-50}$ mutant mice and commercial C57BL/6 WT mice were used for the preparation of MEFs. MEFs were isolated and cultured as previously described[40]. Briefly, Embryos were isolated from E12.5–E13.5 mouse embryos. After the head and most of the internal organs were removed, each embryo was minced and digested for 15 min, and then cultured at 37 °C, 5% CO$_2$ in freshly prepared MEF medium composed DMEM, 10% FBS, 2 mM L-glutamine, and 100 U/ml penicillin-streptomycin. For Ca$^{2+}$-imaging analysis, AAV1 particles (Addgene, Cat # 100836-AAV1) carrying CAG-driven GCaMP6f Ca$^{2+}$ sensor were added in the medium for 16–24 h at 37 °C. Cells were analyzed 24–48 h after a change of the medium to remove the virus. All animal experiments were carried out according to protocols and guidelines approved by the Institutional Animal Care and Use Committee of the University of Texas MD Anderson Cancer Center.

### Imaging analysis of NAADP-evoked Ca$^{2+}$ release.

Transfected cells were identified by fluorescence of the Ca$^{2+}$ reporter GCaMP6f. Fluorescence was monitored with an Axio Observer A1 microscope equipped with an AxioCam MRm digital camera and ZEN Blue 2 software containing a Physiology module (Carl Zeiss) at a sampling frequency of 2 Hz. Cell injection was performed with a FemtoJet microinjector (Eppendorf). The pipette solution contained (mM): 110 KCl, 10 NaCl, and 20 HEPES (pH 7.2) supplemented with Dextran (10,000 MW)-Texas Red (0.3 mg/ml) and NAADP (100 nM) or vehicle. The bath was Hank's

Balanced Salt Solution (HBSS) that contained (mM): 137 NaCl, 5.4 KCl, 0.25 $Na_2HPO_4$, 0.44 $KH_2PO_4$, 1 $MgSO_4$, 1 $MgCl_2$, 10 glucose, and 10 HEPES (pH 7.4). When purified hLsm12-His$_{E.coli}$ was used, it was injected at 100 ng/μl (~4 μM) with NAADP in the same pipette solution. To minimize interference by contaminated $Ca^{2+}$, the pipette solution was always treated with Chelex 100 resin (Cat# C709, Sigma-Aldrich) immediately before use. Microinjection (0.5 s at 150 hPa) was made ~30 s after pipette tip insertion into cells. Only cells that showed no response to mechanical puncture, i.e., no change in GCaMP6f fluorescence for ~30 s, were chosen for pipette solution injection. The successful injection was verified by fluorescence of the co-injected Texas Red. Elevation in intracellular $Ca^{2+}$ concentration was reported by a change in fluorescence intensity $\Delta F/F_0$, calculated from NAADP microinjection-induced maximal changes ($\Delta F$ at the peak) in fluorescence divided by the baseline fluorescence ($F_0$) immediately before microinjection.

**Patch-clamp recording of TPC currents**. To identify transfected cells under a fluorescence microscope for patch-clamp recording, TPC1 or TPC2 was co-expressed with GFP in HEK293 cells via transfection with either a biscistronic vector pCDNA6-TPC2-V5-IRES-AcGFP or the channel's cDNA construct (pCDNA6-TPC1-V5-FLAG or pCDNA6-TPC2-V5-FLAG) together with pEGFP-C1. NAADP-induced TPC activation was recorded by whole-cell patch-clamp recordings on HEK293 cells with asymmetric $Na^+$ (outside)/$K^+$ (inside) solutions using a MultiClamp 700B amplifier and pCLAMP software (Axon Instruments) at room temperature. Bath solution contained 145 mM NaMeSO$_3$, 5 mM NaCl, and 10 mM HEPES (pH 7.2). Pipette electrodes (3–5 MΩ) were filled with 145 mM KMeSO$_3$, 5 mM KCl, and 10 mM HEPES (pH 7.2). The cells were visualized under an infrared differential interference contrast optics microscope (Zeiss). Currents were recorded by voltage ramps from −120 to +120 mV over 400 ms for every 2 s with a holding potential of 0 mV. After a whole-cell recording configuration was achieved, an injection pipette was inserted into the cell and the baseline of the whole-cell current was recorded. Microinjection of NAADP and purified Lsm12 protein was performed as above in imaging analysis of NAADP-evoked $Ca^{2+}$ release. The NAADP-induced currents were obtained by subtraction of the baseline from NAADP injection-induced currents. PI(3,5)P$_2$-activated TPC2 activation was recorded by a patch-clamp recording of whole enlarged endolysosomes as reported[14,41]. Cells were treated with vacuolin (1 μM) overnight to enlarge endo-lysosomes. Patch pipettes for recording were polished and had a resistance of 5–8 MΩ. The cytoplasmic solution contained 160 mM NaCl and 5 mM HEPES (pH was adjusted with NaOH to 7.2). The luminal solution contained 105 mM CaCl$_2$, 5 mM HEPES, and 5 mM MES (pH was adjusted to 4.6 with methanesulfonic acid).

**Statistics and reproducibility**. The data was processed and plotted with Igor Pro (v5), GraphPad Prism (v8), or OriginLab (v2015 or 2017). Unpaired Student's $t$-test (two-tailed) was used to calculate $p$ values. Unless indicated, all measurements or repeats were taken with distinct samples or cells. Independent experiments with similar results related to representative results were done three times for Fig. 1e left panel, four times for Fig. 1e right panel, two times for Fig. 1f left panel, three times for Fig. 1f right panel, three times for Fig. 2a, two times for Fig. 2b, one time for Fig. 2f, one time for Fig. 4d, two times for Fig. 4e, one time for Fig. 4f, two times for Fig. 5a, two times for Fig. 5b, two times for Fig. 5e, one time for Fig. 7d, three times for Fig. 7e, two times for Fig. 7f, one time for Supplementary Fig. 3c, three times for Supplementary Fig. 5a, one time for Supplementary Fig. 5b, and four times for Supplementary Fig. 5c.

**Reporting Summary**. Further information on research design is available in the Nature Research Reporting Summary linked to this article.

## Data availability
The data that support this study are available from the corresponding author upon reasonable request. The raw LC-MS/MS data generated in this study have been deposited in the MassIVE database (https://massive.ucsd.edu) under accession codes MSV000087415, MSV000087416, MSV000087417, and MSV000087418. Source data are provided with this paper.

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

## Acknowledgements

We thank Michael X. Zhu (The University of Texas Health Science Center at Houston) and Youxing Jiang (The University of Texas Southwestern Medical Center) for manuscript reading, discussion, and comments. We thank Meggie Young for assistance in protein purification and all lab members for discussion. Mass spectrometry raw data were collected at the Proteomic Cores of the University of California in Davis and the University of Texas Southwestern Medical Center and Shenzhen Huada Gene Technology Co Ltd. The founder lines of mutant mice were generated at MD Anderson Cancer Center Genetically Engineered Mouse Facility. This work was supported by National Institutes of Health grants NS096296 (J.Y.) and GM130814 (J.Y.).

## Author contributions

J.Z. performed protein expression/characterization/isolation, proteomic analysis, and protein-ligand and protein–protein binding assays and initiated $Ca^{2+}$-imaging and Lsm12-KO experiments. X.G. generated Lsm12-KO cells and mutant mice, performed immunofluorescence, performed microinjection, and collected $Ca^{2+}$-imaging and electrophysiological data. K.S. contributed to protein purification. J.Z. and X.G. generated and characterized Lsm12-KD cells. J.Y. conceived and supervised the project and constructed plasmids. J.Z., X.G., K.S., and J.Y. designed the research, analyzed data, and wrote the manuscript.

## Competing interests

The authors declare no competing interests.
