## [Peer Review File · Nature Communications]

Reviewers' Comments:

Reviewer #1:

Remarks to the Author:

The authors make a highly compelling case for Lsm12 as the long sought-after NAADP receptor that gates TPCs. This is a highly important work that could potentially revolutionize the endo-lysosomal Ca²⁺ signalling field as well as (intriguingly) uniting this field to RNA processing. This deserves to be published in the highest quality journal.

The authors have used a sophisticated multi-pronged approach to substantiate their claims which is entirely appropriate and impressive.

However, there remain some outstanding issues which need to be addressed before publication.

Major Points

1. In terms of Ca²⁺ and currents, NAADP concentration-response is diagnostically and uniquely bell-shaped. Fig S4a is clearly not. This is a substantial concern, and a discrepancy with the field (and even other HEK-cell studies). In their hands, higher [NAADP] is evoking larger Ca²⁺ responses than expected. One possible explanation is that the more concentrated NAADP solutions are contaminated with Ca²⁺ and, therefore, the 'responses' are NOT dependent on NAADP. This can be experimentally addressed in two ways: (a) Are higher concentrations ($\geq 10 \mu\text{M}$ NAADP) inhibited by Ned-19 (they only test Ned-19 with 100nM NAADP; Fig S4b)? (b) The Ca²⁺ contamination can be removed by treating the pipette solution with Chelex resin to mop-up the Ca²⁺. Even if the responses prove to be Ned-19 sensitive, a lack of a bell-shaped curve would still be a worrying difference to the field, and will require a compelling explanation.
2. Lsm12 was selected because it was the only protein in the intersection of TPC1/TPC2/NAADP. The authors should therefore also test whether Lsm12 mediates TPC1 activation by NAADP (Ca²⁺ and currents).
3. There is a significant discrepancy between the NAADP binding affinity of Lsm12 and the Ca²⁺-release. The K_d of binding is 20-30nM. The authors observe maximal Ca²⁺ release with 100nM NAADP in the pipette; assuming a 1% injection volume, the cytosolic [NAADP] will be 1nM (which should barely bind to Lsm12, given the K_d), and so the affinity for Ca²⁺ release will be 0.1 nM (or lower). This is a concern. Can the authors offer an explanation?
4. How specific is Lsm12? Do other Lsm proteins (large or small) substitute (also see point 6)?
5. Does PI(3,5)P₂ evoke TPC2 currents in Lsm12-KO cells? Lsm12-KO should only affect NAADP and not PI(3,5)P₂ if it is a specific NAADP receptor and not just a general TPC2 modulator (see Gerndt 2020).
6. The authors use ATP as a control for the selectivity of the effect of Lsm12 KO which is a good idea. Unfortunately, they used a maximal concentration (50 μM) when really they should have used a sub-maximal concentration (e.g. 0.3 μM); this might mask Lsm12-KO effects.
7. In experiments co-injecting purified Lsm12 protein, there should be some additional controls: (i) heat-inactivated Lsm12; (ii) another Lsm protein (small or large).
8. The authors do not consistently perform [³²P]NAADP binding in the same preparation. Sometimes they use purified bacterial His-Lsm12, in others, they use Lsm12-KO HEK cells. Does this affect the binding (e.g. affinity)? This might resolve the affinity discrepancy (Point 3). For example, Lsm12 protein might be post-translationally modified in HEKs but not in bacteria which might affect function.
9. I am concerned by the 'novel' and slightly bizarre methodology of the electrophysiology protocols.
 - a. With TPC2-LLAA, why didn't the authors simply adopt a straightforward whole-cell (WC) protocol with NAADP in the patch pipette (which is less invasive and will be required anyway for the new PI(3,5)P₂ controls – point 4)? The mechanical disruption of injection will surely make artifacts more likely during WC recordings.
 - b. Several electrophysiology approaches lose NAADP responsivity (e.g. excised plasma membrane patches, enlarged endo-lysosomal patching) and I also wonder if the authors resorted to NAADP injection because of a loss of response due to dialysis of the cytosol (and loss of endogenous Lsm12). Each of these 'broken systems' might be rescued by restoring the lost Lsm12 (either by including purified protein in the patch-pipette/bath or by using the TPC2-Lsm12 fusion). It is very important for the authors to address this as it has plagued the field for years and they might solve the long-standing

inconsistencies.

Minor Points

- a) The complete amino acid and DNA sequences (and Accession numbers) of Lsm12 must be detailed, as should the sequence-definitions of their Lsm-, AD- and Linker-domains.
- b) They must show that all the TPC constructs target to the lysosome (particularly the large Lsm12-fusion protein). Pearson's correlation coefficients should be determined with a lysosomal marker. Similarly, I would expect to see some Lsm12 on endo-lysosomes (small puncta on a cytosolic background), even if only with TPC over-expression). Can the authors show this?
- c) In Fig 5, indicate in the Legend how long after NAADP injection the voltage ramps were determined.
- d) The authors should refer to the recent Gerndt 2020 eLife paper, particularly relevant for the additional PI(3,5)P2 controls.
- e) They find that Lsm12-KO does not alter TPC over-expression, but the more relevant issue is whether Lsm12 expression/activity alters endogenous TPC levels e.g. in HEKs, MEFs.

Reviewer #2:

Remarks to the Author:

Title: Lsm12 is an NAADP receptor and a two-pore channel regulatory protein required for calcium mobilization from acidic organelles

Authors: Jiyuan Zhang, Xin Guan, and Jiusheng Yan

Comments:

The manuscript by Zhang and co-authors describes an effort to characterize and identify Lsm12 protein as the regulatory protein for Ca mobilization, via interacting with both NAADP and TCP1/2.

My concern in this manuscript is the whole set of experiments that were done to identify Lsm12 as a protein of importance for further experiments. I hope the authors will find my comments useful to better represent their data and findings related to mass spectrometry (Fig 1).

1) While I really appreciate the illustrations in Fig. 1a, I believe it could be presented in a more clear way, perhaps highlighting within the scheme where do we expect 'NAADP interacting proteins' to elute. The similar would be for TPC1 and TPC2, and these should be separated, so it is clear. Also, I am missing a detail in the illustration after TPC transfection, it looks like after AP with anti-FLAG eluates look the same from the figure. That should be corrected.

2) I am not seeing much details in Materials section about how SILAC experiment was done. Authors didn't say how much of heavy and light-labeled proteins they mixed for the experiment

3) Authors should explain better their in-gel digestion procedure. If they cut gels in 3 sections, they should explain why and what was done later. I assume the gels were combined, but it should be stated. Also, I would like to know which trypsin was used for this experiment, and how peptides were extracted from the gel.

4) I would like to know if any protease inhibitors were used for MS experiment

5) I would like to see raw MS data files

6) Authors should explain in how many replicated experiments were done, for both affinity purification and MS part.

7) Authors should give more details about bioinformatics part for MS experiment. I am missing which database was used, details for FDR and similar.

8) I would like to see uncropped western blots for Fig. 1e and 1f. Also, I would like to know in how many replicates this experiment was done.

Reviewer #3:

Remarks to the Author:

The paper by Zhang et al. identifies Lsm-12 as a NAADP-binding protein proposed to confer NAADP sensitivity to two-pore channels (TPCs). Identification of how TPCs respond to NAADP has been a long standing unknown in this field, as no direct binding site for NAADP has been identified on the TPC itself. Identification of Lsm-12 as a NAADP receptor is potentially a highly significant finding for the field. While there is strong enthusiasm for both the methodological scope of this study and potential impact of this discovery, there are several areas requiring additional data and/or clarification.

Major

#1. Apart from single cell traces in Figure 2b, no raw data of calcium signals are presented from any of the manipulations described throughout in the manuscript (including all SKBR3 and MEF data). Trace kinetics for all of the key calcium imaging experiments need to be provided for Figures 3c/e & Figure 4 & Figure 6 & Figure S4.

#2. A key claim made by the authors is that LSM12 is a universal NAADP-BP that couples NAADP binding to regulation of both TPC1 and TPC2. However very little data for TPC1 is provided. Additionally, most data shown are dependent on (TPC2) overexpression. Both these issues require elaboration.

First, functional data with TPC1 overexpression and rescue in the KO line should be provided (Figure 2). Binding data for TPC1, as shown for TPC2 (Figure 3 & S5), would provide support for the authors' assertion that LSM12 regulates NAADP-sensitivity of both TPC isoforms.

Second, does siRNA of LSM12 decrease the endogenous NAADP-evoked Ca²⁺ signal in MEFs (Figure 6A)?

#4. Missing control. A NAADP-binding defective construct (e.g. Delta-Lsm) should be used as a negative control for the protein injection experiments (Figure 2b).

#5. Missing control. siRNA of any of the other 'long' LSM proteins (11, 14A/B, 16) should be added as a control for the calcium imaging assays.

#6. Overexpression of the epitope tagged Lsm12 (Figure 2a, lane 3) yields a weaker immunoreactive signal than endogenous LSM12, suggesting that the KO clone has poor generalized transfectability as also seen in immunofluorescence data (Figure 2a right). I therefore have some skepticism as to data in triple transfected KO cells (reporter, TPC2, LSM12) as to the relative expression levels of these heterologously expressed constructs, variation in any of which could impact the sole provided DF/F readout. Please clarify whether ATP controls (Figure S4) have been collected in the same cells where the functional effects of LSM manipulation are being reported.

#7. Again in the context of data being dependent on overexpression, the pipeline for identification of the NAADP interacting proteins depends on overexpression of GFP/FLAG-tagged TPC isoforms, prior to isolation of NAADP binding proteins. Mass spec datasets for lysates passed over the NAADP affinity resin should be presented in the absence of TPC overexpression. From the authors own dataset (Fig 1c) overexpression of TPC isoforms significantly and uniquely alters the NAADP interactome (and perhaps the endogenous TPC interactome), so the lack of/identity of the endogenous background used

in these studies is unclear.

Minor

#1. The text is imprecise when referring to construct expression. Please refer to the exact construct expressed throughout the manuscript (e.g. TPC1-eGFP-FLAG expressing and not simply 'TPC1 expressing'). Experimental conditions are also poorly described – for example, are TPCs expressed in the functional assays in Figure 4? Similarly, please confirm the MEF experiments (Figure 6A) are in the absence of TPC overexpression.

#2. The binding selectivity of recombinant LSM12 for NAADP over NADP is remarkable (no effect of NADP up to 100uM), and more selective than ever reported in a physiological background. This deserves comment.

#3. Other TPC proteomic datasets have been published over the last decade. Has LSM12 been identified previously?

#4. Do the authors have insight as to whether TPC-FLAG proteomics resemble TPC-FLAG-GFP proteomics, given the potential for non-specific associations with GFP?

RESPONSES TO REVIEWER COMMENTS

We appreciate the reviewers' thorough reading of the manuscript and encouraging, thoughtful, and constructive comments to improve this work. We have taken the comments and concerns seriously and tried our best to address them within the scope of this work. Please note that we have thoroughly revised this manuscript (reorganized text and figures) to improve its readability.

Reviewer #1 (Remarks to the Author):

The authors make a highly compelling case for Lsm12 as the long sought-after NAADP receptor that gates TPCs. This is a highly important work that could potentially revolutionize the endo-lysosomal Ca²⁺ signalling field as well as (intriguingly) uniting this field to RNA processing. This deserves to be published in the highest quality journal. The authors have used a sophisticated multi-pronged approach to substantiate their claims which is entirely appropriate and impressive.

We greatly appreciate the reviewer's positive evaluation of this work's significance and quality. We feel it is necessary to validate a new protein function, in this case the Lsm12 as a critical NAADP receptor, from multiple angles with multidisciplinary approaches despite the need of many years of hard research work.

However, there remain some outstanding issues which need to be addressed before publication.

Major Points

1. In terms of Ca²⁺ and currents, NAADP concentration-response is diagnostically and uniquely bell-shaped. Fig S4a is clearly not. This is a substantial concern, and a discrepancy with the field (and even other HEK-cell studies). In their hands, higher [NAADP] is evoking larger Ca²⁺ responses than expected. One possible explanation is that the more concentrated NAADP solutions are contaminated with Ca²⁺ and, therefore, the 'responses' are NOT dependent on NAADP. This can be experimentally addressed in two ways: (a) Are higher concentrations ($\geq 10 \mu\text{M}$ NAADP) inhibited by Ned-19 (they only test Ned-19 with 100nM NAADP; Fig S4b)? (b) The Ca²⁺ contamination can be removed by treating the pipette solution with Chelex resin to mop-up the Ca²⁺. Even if the responses prove to be Ned-19 sensitive, a lack of a bell-shaped curve would still be a worrying difference to the field, and will require a compelling explanation.

The pipette solution was always treated with Chelex resin first before use. We redid the experiments to include Ned-19 and no TPC transfected cells as negative controls. We revised this part in page 5 as "A bell-shaped dose response, which was peaked at 100 nM NAADP (pipette solution), was observed when low concentrations of NAADP in pipette solution were injected (Fig. S4a). When higher concentrations (1 and 10 μM in pipette solution) of NAADP were used, non-specific response was increased as some responses remained in the absence of exogenous TPC expression or in the presence of cell pretreatment with trans-Ned-19 (10 μM), an antagonist of NAADP-mediated Ca²⁺ release". What important is the low concentration of the NAADP we used throughout the study. Under this condition, we have multiple negative controls data and the data are consistent with what were reported in the field.

2. Lsm12 was selected because it was the only protein in the intersection of TPC1/TPC2/NAADP. The authors should therefore also test whether Lsm12 mediates TPC1 activation by NAADP (Ca²⁺ and currents).

We included TPC1 data in Ca²⁺ imaging (Fig. 2e) and whole cell current recording (Fig. 6d) to show that Lsm12 also mediates NAADP-evoked TPC1 activation and Ca²⁺ elevation in TPC1-expressing cells.

3. There is a significant discrepancy between the NAADP binding affinity of Lsm12 and the Ca²⁺-release. The K_d of binding is 20-30nM. The authors observe maximal Ca²⁺ release with 100nM NAADP in the pipette; assuming a 1% injection volume, the cytosolic [NAADP] will be 1nM (which should barely bind to Lsm12, given the K_d), and so the affinity for Ca²⁺ release will be 0.1 nM (or lower). This is a concern. Can the authors offer an explanation?

We didn't know exactly how much volume of NAADP solution was injected. As we could see the volume increase under microscope upon injection, we estimate that it could be between 1-10%. In addition, upon microinjection the local concentration of NAADP could be high which might trigger and then propagate the Ca²⁺ elevation inside the cells.

4. How specific is Lsm12? Do other Lsm proteins (large or small) substitute (also see point 6)?

It is possible that other Lsm protein might function in NAADP signaling. However, we prefer not to go much beyond Lsm12 in this manuscript for the following reasons: 1) we only identified Lsm12 (no other Lsm protein) in our quantitative proteomic analysis; 2) Lsm12 KO fully eliminates the NAADP response; 3) we added new data (Fig. S4d-f) to show that knockdown of 2 tested other Lsm proteins, Lsm5 and Lsm11, had much smaller effects than knockdown of Lsm12 on the cells' response to NAADP. We chose these 2 other Lsm proteins for the knockdown experiment in this revision was because they are a short and long form of Lsm protein, respectively, and because the predesigned RT PCR primers that we obtained commercially worked well for detecting RNA level of these 2 proteins. Therefore, from these results we tend to think Lsm12 is a dominant NAADP receptor in HEK293 cells. The other important consideration that we don't want to study further about other Lsm proteins in this revision is as such. We think that extensive and systematic studies are needed to either disprove or prove that other or some other Lsm proteins also function as NAADP receptor. For example, even an Lsm protein doesn't function in NAADP signaling in HEK 293 cells, it is still possible that it can function in NAADP signaling in other cells as its expression level and biochemical state could vary. On the other hand, it will require a comprehensive work of multiple approaches, as we did for Lsm12, to prove that another Lsm protein can function as an NAADP binding protein in NAADP signaling. Therefore, we leave this question as future work and we are neutral in discussion in this revision to neither support nor against the possibility of other Lsm proteins in NAADP signaling.

5. Does PI(3,5)P₂ evoke TPC2 currents in Lsm12-KO cells? Lsm12-KO should only affect NAADP and not PI(3,5)P₂ if it is a specific NAADP receptor and not just a general TPC2 modulator (see Gerndt 2020).

We added data on dose response of PI(3,5)P₂ on TPC2 activation with WT and Lsm12-KO cells (Fig. S6a). We did this via patch clamp recording of whole enlarged lysosomes. We found no significant difference in the channels' sensitivity to PI(3,5)P₂ with/without Lsm12 expression. However, we are cautious in interpreting the data as we don't know whether Lsm12 remained significantly associated with TPC2 under those invasive recording conditions. If Lsm12 can directly interact with the channels, it could still potentially affect some other gating properties of TPC. For whole cell recording, we were unable to record good PI(3,5)P₂ microinjection induced whole cell TPC2 currents. Presumably, the injected PI(3,5)P₂ might be hard to reach to plasma membrane under this condition, e.g., PI(3,5)P₂ is absorbed by endolysosomal membrane system before reaching plasma membrane.

6. The authors use ATP as a control for the selectivity of the effect of Lsm12 KO which is a good idea. Unfortunately, they used a maximal concentration (50μM) when really they should have used a sub-maximal concentration (e.g. 0.3 μM); this might mask Lsm12-KO effects.

We agree that the used 50 μM ATP concentration is high. We now replace that with new data of 1 μM ATP (Fig. S4b) and observed no significant difference between WT and KO cells for 1 μM ATP induced Ca^{2+} increase in HEK293 cells. The cells for both WT and Lsm12-KO cells were transiently transfected with TPC2 as in most other Ca^{2+} imaging experiments.

7. In experiments co-injecting purified Lsm12 protein, there should be some additional controls: (i) heat-inactivated Lsm12; (ii) another Lsm protein (small or large).

We added heat-inactivated Lsm12 as negative control (Fig. 2d and Fig. 6c). We didn't add other Lsm proteins because it is beyond our focus on Lsm12 in this manuscript and we don't have readily available other Lsm proteins to use.

8. The authors do not consistently perform $[^{32}\text{P}]\text{NAADP}$ binding in the same preparation. Sometimes they use purified bacterial His-Lsm12, in others, they use Lsm12-KO HEK cells. Does this affect the binding (e.g. affinity)? This might resolve the affinity discrepancy (Point 3). For example, Lsm12 protein might be post-translationally modified in HEKs but not in bacteria which might affect function.

It is likely that the NAADP binding property of purified His-Lsm12 expressed in E. coli is slightly different from that in HEK cells as Lsm12 in HEK cells may have modifications and interacting proteins. However, it is difficult to get pure preparation of Lsm12 protein expressed in HEK cells. Thus, in order to draw a solid conclusion that Lsm12 is an NAADP binding protein, we preferred to use the purified His-Lsm12 expressed in E. coli for the binding assay as it is pure without contamination or interactions by other mammalian cell proteins. Now, we added data on NAADP binding on endogenous Lsm12 in HEK293 cells using immobilized NAADP to pull down endogenous Lsm12 in HEK cells and free NAADP to compete in binding (Fig. 3c and Fig. S5c). The result for NAADP binding is similar to what we got with purified His-Lsm12 expressed in E. coli. Because of limitation in experimental condition, the results from in vitro binding assays might not fully represent what occurs inside cells. However, as it is nearly impossible to measure directly inside cells, it is difficult to know what is the receptors' K_d for NAADP inside cells. For example, the injected NAADP could work at a high local concentration (before diffusion) to trigger Ca^{2+} release locally and then propagate the Ca^{2+} release in whole cell or it could act globally at a lower concentration after diffusion inside cells.

9. I am concerned by the 'novel' and slightly bizarre methodology of the electrophysiology protocols.

a. With TPC2-LLAA, why didn't the authors simply adopt a straightforward whole-cell (WC) protocol with NAADP in the patch pipette (which is less invasive and will be required anyway for the new $\text{PI}(3,5)\text{P}_2$ controls – point 4)? The mechanical disruption of injection will surely make artifacts more likely during WC recordings.

To ensure that what we observed were not artifacts, we employed multiple negative controls, no NAADP, no transfection, TPC-WT (not targeted to plasma membrane), TPC pore mutant L265P, and now heat-inactivated Lsm12 to rule out this artifact possibility. For us, it is difficult to get reliable NAADP response by adding NAADP in pipette solution in whole cell recording because it is hard if not impossible to get reliable baseline (no NAADP) for subtraction of the leak currents. We preferred to use the microinjection method because it is close to the condition used for NAADP-microinjection induced Ca^{2+} release in our Ca^{2+} imaging experiments. We consider this method 'novel' or 'unconventional' in term of drug delivery in whole cell recording which has certain advantages that allows obtaining baseline without drug and a click-speed drug delivery directly inside cells. This could be particularly useful if the response to a drug is small or transient which might not be observed well when the drug is diffused to the cell via a recording pipette. The property of NAADP-induced TPC activation is not well established and characterized. We felt it

was safer to apply NAADP in this way. We now removed the word “novel” and explained the reasons to use this method in discussion (page 13).

b. Several electrophysiology approaches lose NAADP responsivity (e.g. excised plasma membrane patches, enlarged endo-lysosomal patching) and I also wonder if the authors resorted to NAADP injection because of a loss of response due to dialysis of the cytosol (and loss of endogenous Lsm12). Each of these ‘broken systems’ might be rescued by restoring the lost Lsm12 (either by including purified protein in the patch-pipette/bath or by using the TPC2-Lsm12 fusion). It is very important for the authors to address this as it has plagued the field for years and they might solve the long-standing inconsistencies.

The reviewer raised an intriguing question which has long been a puzzle. The identification of NAADP binding proteins is expected to help address this question. Part of the reason that it took long time for this revision is because we had spent significant efforts doing both inside-out and whole lysosome patch-clamp recordings to address this question during the revision period. We had tried different recording conditions. So far, we haven’t been able to see any NAADP-induced TPC2 activation by perfusion of Lsm12-NAADP to the inside-out plasma membrane patches. For whole lysosomal recording, the TPC2 channel’ response to NAADP occurred only in a portion of recorded lysosomes despite its dependence on Lsm12. These preliminary results suggest that a more stringent condition, which naturally occurs inside cells but hard to retain outside of cells, might need to meet for NAADP-induced TPC activation. We are very interested in solving this problem. However, we realized that the new data we obtained is too preliminary to include and it is going to take significant more systematic study and effort to identify the missing condition or factor to robustly reproduce the NAADP-induced TPC activation in vitro (inside-out or lysosomal recording). We agree that solving this puzzle is important. But we feel that it is neither the focus and nor essential for this manuscript as we already demonstrated with whole cell patch-clamp recording that NAADP by microinjection can activate TPC2 in a strictly Lsm12-dependent manner, a recording and NAADP delivery condition that is close to our Ca²⁺ imaging assay of NAADP-evoked Ca²⁺ elevation. Therefore, we decide to leave this question to address in future.

Minor Points

a) The complete amino acid and DNA sequences (and Accession numbers) of Lsm12 must be detailed, as should the sequence-definitions of their Lsm-, AD- and Linker-domains.

We added this information in the text and also in figure legend (now Fig. 4a).

b) They must show that all the TPC constructs target to the lysosome (particularly the large Lsm12-fusion protein). Pearson’s correlation coefficients should be determined with a lysosomal marker. Similarly, I would expect to see some Lsm12 on endo-lysosomes (small puncta on a cytosolic background), even if only with TPC over-expression). Can the authors show this?

We didn’t do experiments on cellular localization of TPC constructs because it had been well established that the channels only or mainly express on endolysosomal membranes. We now added Duo-Link data (Fig. 5e) to show that Lsm12 and TPC2 do colocalize. For the TPC2-Lsm12 fusion, we chose to remove data from this construct as it doesn’t occur naturally, and it is redundant to include.

c) In Fig 5, indicate in the Legend how long after NAADP injection the voltage ramps were determined.

The voltage ramps were given continuously to monitor the currents before and after the injection. The data shown is approximately 2 seconds (1 sweep immediately) after injection when the NAADP induced currents are maximal. The currents gradually became smaller presumably because of dilution caused by diffusion of NAADP to pipette solution.

d) The authors should refer to the recent Gerndt 2020 eLife paper, particularly relevant for the additional PI(3,5)P2 controls.

Thanks for reminding. We cited this paper in the revised manuscript.

e) They find that Lsm12-KO does not alter TPC over-expression, but the more relevant issue is whether Lsm12 expression/activity alters endogenous TPC levels e.g. in HEKs, MEFs.

We paid attention to over-expressed TPC in HEK cells is because the endogenous TPC level is too low to be able to give the NAADP-induced Ca²⁺ release. So, we consider the endogenous TPC expression in HEK cells is not relevant in this study as the experiments were always done with TPC overexpression. For MEFs, the Lsm12 mutant used is only a 6-residue loop deletion mutation. We didn't expect a major impact on expression of other proteins, e.g., TPCs. Importantly, we validated in HEK293 cells that this deletion mutation is defective in NAADP-evoked Ca²⁺ release and TPC2 activation. So, it should not be related to changes in TPC expression in MEFs.

Reviewer #2 (Remarks to the Author):

Comments:

The manuscript by Zhang and co-authors describes an effort to characterize and identify Lsm12 protein as the regulatory protein for Ca mobilization, via interacting with both NAADP and TCP1/2.

My concern in this manuscript is the whole set of experiments that were done to identify Lsm12 as a protein of importance for further experiments. I hope the authors will find my comments useful to better represent their data and findings related to mass spectrometry (Fig 1).

We thank the reviewer's constructive comments on the biochemical part of this work. As this biochemical part serves a main purpose to identify the potential target protein of NAADP, we didn't emphasize in detail on this part in last version of this manuscript. Now, we have added more details and revised some description to improve clarity. We hope this revised version will convey information much better.

1) While I really appreciate the illustrations in Fig. 1a, I believe it could be presented in a more clear way, perhaps highlighting within the scheme where do we expect 'NAADP interacting proteins' to elute. The similar would be for TPC1 and TPC2, and these should be separated, so it is clear. Also, I am missing a detail in the illustration after TPC transfection, it looks like after AP with anti-FLAG eluates look the same from the figure. That should be corrected.

We are sorry for the confusion. We have revised Fig. 1a to make it better to understand by adding some more detail.

2) I am not seeing much details in Materials section about how SILAC experiment was done. Authors didn't say how much of heavy and light-labeled proteins they mixed for the experiment

We added this information to the method. Typically, for each replicate of one SILAC experiment, 5 dishes of heavy/light cells were used for the pulldown. Depending on the cell density, approximately 5 mg of heavy/light labelled proteins was extracted from the cells, and volumes of the cell lysate were slightly adjusted with the lysis buffer to ensure both the protein amount and concentration were the same between heavy and light labeled proteins before the pulldown experiment.

3) Authors should explain better their in-gel digestion procedure. If they cut gels in 3 sections, they should explain why and what was done later. I assume the gels were combined, but it should be stated. Also, I would like to know which trypsin was used for this experiment, and how peptides were extracted from the gel.

We added some description about in-gel digestion including the trypsin information in the method. We revised the method to clarify the procedure. The pulldown eluates from heavy and light labeled samples were pooled before SDS-PAGE. The combined samples were run into an SDS-PAGE gel, cut into 3 sections (for NAADP pulldown) or 5 sections (for anti-FLAG pulldown) and proceeded to in-gel digestion separately. Generated peptides were also injected into mass spectrometry separately. The reason we used this strategy is that from our experience, it can generate more detectable peptide spectra for mass spectrometry compared with combining samples into one fraction, while avoiding over diluting the samples if separated into more fractions (such as 10) due to the relatively less complexity of the pulldown product (than, for example, total cell lysate), which will decrease the number of identifiable spectra.

4) I would like to know if any protease inhibitors were used for MS experiment

Yes, the protease inhibitor cocktail from Roche Inc. (cOmplete, EDTA free) was used in lysis buffer throughout the affinity purification process. We added this piece of information in the method.

5) I would like to see raw MS data files

The raw data for the 4 sets of the protein interactomes are now available massive.ucsd.edu for download (password: b9f5e3r5) at a public server via ftp://MSV000087415@massive.ucsd.edu , ftp://MSV000087416@massive.ucsd.edu, ftp://MSV000087417@massive.ucsd.edu , ftp://MSV000087418@massive.ucsd.edu . We found the data can be better accessed via an FTP program, e.g., WinSCP, with the following setting:

File protocol: FTP

Encryption: No encryption

Host name: massive.ucsd.edu

Port number: 21

User name: MSV000087415 (or other 3 user names above)

Password: b9f5e3r5

6) Authors should explain in how many replicated experiments were done, for both affinity purification and MS part.

We added this information in the method. For anti-FLAG pulldown experiments, 5 complete replications (starting from cell culture) were performed on TPC1 and TPC2, respectively. For NAADP pulldown, however, we did the experiment for only once. We didn't do more replication after we found LSM12. For peptide samples, only 1 set of samples (TPC2 interactome replicate 1) were reanalyzed by MS.

7) Authors should give more details about bioinformatics part for MS experiment. I am missing which database was used, details for FDR and similar.

We added those to the method as advised.

8) I would like to see uncropped western blots for Fig. 1e and 1f. Also, I would like to know in how many replicates this experiment was done.

The raw western blots (film data) are now included in the source data. We also included additional raw film data (a separate excel file of additional data) for the repeats or related experimental results (e.g., pulldown in reverse direction) that are not shown in figures. In last version, we made an error for Fig. 1e (bottom/right) with a wrong western blot image and the problem is now fixed. The experiments related to Fig. 1e (left panels) was done 3 times: once for pulldown of Lsm12 by TPC1 and twice for reverse direction (pulldown of TPC1 by Lsm12). The experiments related to Fig. 1e (right panels) was done 4 times: twice for pulldown of Lsm12 by TPC2 and twice for reverse direction (pulldown of TPC2 by Lsm12). The experiment for Fig. 1f (left panels) was done twice for pulldown of Lsm12 by NAADP in the presence of TPC1 (Fig. 5b left panels is considered as a repeat for Fig. 1f left panels). The experiment for Fig. 1f (right panels) was done 3 times for pulldown of Lsm12 by NAADP in the presence of TPC2 (Fig. 5b right panels is considered as a repeat for Fig. 1f right panels).

Reviewer #3 (Remarks to the Author):

The paper by Zhang et al. identifies Lsm-12 as a NAADP-binding protein proposed to confer NAADP sensitivity to two-pore channels (TPCs). Identification of how TPCs respond to NAADP has been a long standing unknown in this field, as no direct binding site for NAADP has been identified on the TPC itself. Identification of Lsm-12 as a NAADP receptor is potentially a highly significant finding for the field. While there is strong enthusiasm for both the methodological scope of this study and potential impact of this discovery, there are several areas requiring additional data and/or clarification.

We feel grateful for the reviewer's thorough reading and constructive assessment and comments of this work. We also appreciate the reviewer's view of the potential high significance of our findings. We agree that additional data/clarification will make the work better.

Major

#1. Apart from single cell traces in Figure 2b, no raw data of calcium signals are presented from any of the manipulations described throughout in the manuscript (including all SKBR3 and MEF data). Trace kinetics for all of the key calcium imaging experiments need to be provided for Figures 3c/e & Figure 4 & Figure 6 & Figure S4.

We added representative traces in the supplemental data (Fig. S4c).

#2. A key claim made by the authors is that LSM12 is a universal NAADP-BP that couples NAADP binding to regulation of both TPC1 and TPC2. However very little data for TPC1 is provided. Additionally, most data shown are dependent on

(TPC2) overexpression. Both these issues require elaboration.

First, functional data with TPC1 overexpression and rescue in the KO line should be provided (Figure 2). Binding data for TPC1, as shown for TPC2 (Figure 3 & S5), would provide support for the authors' assertion that LSM12 regulates NAADP-sensitivity of both TPC isoforms.

We now added data on TPC1 (Fig. 2e and 6d).

Second, does siRNA of LSM12 decrease the endogenous NAADP-evoked Ca²⁺ signal in MEFs (Figure 6A)?

MEFs were difficult to transfect in our hands. Thus, we didn't do Lsm12 knockdown. Our readily available Lsm12-mutant (Lsm12^{Δ45-50}) MEFs serves the same purpose as siRNA KD. It gives a good loss-of-function (rather than loss-of-expression in KD) control to demonstrate the involvement of Lsm12 in NAADP-induced Ca²⁺ release in MEFs. Given that the mutant has deletion of only a few residues, it could produce less side or global effect unrelated to NAADP signaling than the siRNA KD did. Importantly, we confirmed this mutant's defect in NAADP signaling in HEK293 cells using exogenous recombinant Lsm12 mutant construct. We have no doubt that siRNA KD of Lsm12 will produce similar effect and same conclusion. We feel that the extra work of siRNA KD will add not much additional value in spite of the difficulty in experiment which requires the use of of Lsm12 KD virus.

#4. Missing control. A NAADP-binding defective construct (e.g. Delta-Lsm) should be used as a negative control for the protein injection experiments (Figure 2b).

We added heat-inactivated Lsm12 protein as a negative control in both Ca²⁺ imaging (Fig. 2d) and whole cell patch-clamp recording data (Fig. 6c).

#5. Missing control. siRNA of any of the other 'long' LSM proteins (11, 14A/B, 16) should be added as a control for the calcium imaging assays.

We added new data (Fig. S4d-f) to show that knockdown of 2 tested other Lsm proteins, the short Lsm5 and the long Lsm11, had much smaller effects than knockdown of Lsm12 on the cells' response to NAADP. We chose these 2 other Lsm proteins for the knockdown experiment in this work is because they are a short and long form of Lsm protein, respectively, and because the predesigned RT PCR primers that we obtained commercially worked well for detecting RNA level of these 2 proteins. Same as our answers to another question above. It remains possible that some other Lsm protein might function in NAADP signaling. However, we prefer not to go much beyond Lsm12 in this manuscript.

#6. Overexpression of the epitope tagged Lsm12 (Figure 2a, lane 3) yields a weaker immunoreactive signal than endogenous LSM12, suggesting that the KO clone has poor generalized transfectability as also seen in immunofluorescence data (Figure 2a right). I therefore have some skepticism as to data in triple transfected KO cells (reporter, TPC2, LSM12) as to the relative expression levels of these heterologously expressed constructs, variation in any of which could impact the sole provided DF/F readout. Please clarify whether ATP controls (Figure S4) have been collected in the same cells where the functional effects of LSM manipulation are being reported.

We did notice that the *Lsm12*-KO cells were a little harder to transfect than the WT cells. However, this has not been an issue as when we did Ca^{2+} imaging experiments we chose cells having similar reporter fluorescence. Under overexpression condition for *Lsm12* and TPC2, the expressed proteins are likely more than enough for NAADP-evoked Ca^{2+} release. The ATP controls (new Figure S4b) were done on the same cells transfected with calcium-indicator protein and TPC2.

#7. Again in the context of data being dependent on overexpression, the pipeline for identification of the NAADP interacting proteins depends on overexpression of GFP/FLAG-tagged TPC isoforms, prior to isolation of NAADP binding proteins. Mass spec datasets for lysates passed over the NAADP affinity resin should be presented in the absence of TPC overexpression. From the authors own dataset (Fig 1c) overexpression of TPC isoforms significantly and uniquely alters the NAADP interactome (and perhaps the endogenous TPC interactome), so the lack of/identity of the endogenous background used in these studies is unclear.

We performed the proteomic identification of NAADP interacting proteins in the presence of TPC overexpression is because the NAADP-evoked Ca^{2+} release can only be observed under TPC1 or TPC2 overexpression condition. There are 17 proteins mutually identified in the TPC1-expressing and TPC2-expressing NAADP interactomes, which could be endogenous background. The presence of a great number of proteins that were different between the NAADP interactome with TPC1 expression and that with TPC2 expression is probably because of experimental limitation. To eliminate most false positive proteins, we have done 5 complete replications (starting from cell culture) for identification of TPC1 and TPC2 interactomes. For NAADP interactome with TPC1 and TPC2 expression, however, we did the affinity purification and MS analysis for only once. We didn't do more replication after we found *LSM12*. This might give more variation in the identified proteins between these 2 sets of data.

Minor

#1. The text is imprecise when referring to construct expression. Please refer to the exact construct expressed throughout the manuscript (e.g. TPC1-eGFP-FLAG expressing and not simply 'TPC1 expressing'). Experimental conditions are also poorly described – for example, are TPCs expressed in the functional assays in Figure 4? Similarly, please confirm the MEF experiments (Figure 6A) are in the absence of TPC overexpression.

We now added more plasmid information in the method and also in the results. The MEF experiments were in the absence of TPC overexpression. Only Ca^{2+} indicator protein was transfected to MEFs using virus as it is hard to transfect with plasmids.

#2. The binding selectivity of recombinant *LSM12* for NAADP over NADP is remarkable (no effect of NADP up to 100uM), and more selective than ever reported in a physiological background. This deserves comment.

We added data on NAADP binding on endogenous *Lsm12* in HEK293 cells using immobilized NAADP to pull down endogenous *Lsm12* in HEK cells and free NAADP to compete in binding (new Fig. 3c). The result for NAADP binding is similar to what we got with purified His-*Lsm12* expressed in *E. coli*. However, the selectivity is decreased as NADP can also bind at a high concentration, which is similar to others reported. The reason for the difference in specificity between purified His-*Lsm12* expressed in *E. coli* and endogenous *Lsm12* in HEK cells is unclear. This could be due to some posttranslational modification of *Lsm12* or likely some *Lsm12* interacting protein in HEK cells that either enhance *Lsm12*'s binding to NADP or itself can bind both NADP and NAADP. We overexpressed TPC2 and used detergent to extract the proteins for the binding assay in this experiment, which is different from the binding assay done with purified His-*Lsm12* expressed in *E. coli*.

#3. Other TPC proteomic datasets have been published over the last decade. Has LSM12 been identified previously?

Lsm12 has not been reported in published TPC proteomic datasets. In our case, to ensure elimination of most contaminating proteins and also to be comprehensive in inclusion of interacting proteins, we not only used SILAC-based quantitative proteomics but also repeated many (5) times. We also chose a mild detergent to extract the proteins.

#4. Do the authors have insight as to whether TPC-FLAG proteomics resemble TPC-FLAG-GFP proteomics, given the potential for non-specific associations with GFP?

This is not a major concern in this study as we used GFP-FLAG as the control for TPC-GFP-FLAG. The proteins non-specifically associated with GFP would be present in both the heavy and light samples and displayed as non-differential proteins in the identified proteins.

Reviewers' Comments:

Reviewer #1:

Remarks to the Author:

The authors have very carefully considered my points and addressed them fully and in a constructive manner. The new data added is impressive and solidifies the claims made in the previous version. This work is a major advance to the field.

Minor points

1. L55 I think Ref 14 somewhat addresses the ion permeability problem, and could be cited as such.
2. L56 IPs of TPCs are associated with high affinity NAADP binding Ref 27
3. L195 change "exam" to "examine"
4. L280 change "bindings" to "binding"
5. In the discussion the reduced selectivity of NAADP/NADP should be discussed fully in relation to the exquisite selectivity of recombinant Lsm12 for NAADP over NADP.
6. JTP2 has recently been suggested to also act as an NAADP binding protein in 2 papers in Science Signaling a couple of months back. This should be discussed in the context of this (much for convincing and rigorous) study

Antony Galione

Reviewer #2:

Remarks to the Author:

The authors have significantly improved the manuscript presentation, and addressed all of my concerns and comments from the initial submission. I am therefore satisfied with the quality of data and conclusions the authors have made and presented in this new version. Very promising data and results overall.

Reviewer #3:

Remarks to the Author:

The additional data added in response to the original review has addressed many of the prior concerns. The final study is comprehensive in scope and well executed. Overall this paper will be of exceedingly high impact for the calcium signaling field.

Two comments.

- (1) Papers on HN1L /JPT as a candidate NAADP receptor should be cited given their recent publication.
- (2) Figure 3c. The text should state the measured IC50s for NAADP and NADP displacement, consistent with Panels 3a and 3b.
- (3) Unclear about the methods for new Fig 3c which the authors describe as 'HEK293 endogenous LSm12 pulled down by immobilized NAADP'. If this is simply probing HEK293 lysates there is no real support for the statement that this preparation is just LSM12, as indeed the different binding results may suggest. The statement that this preparation represents endogenous Lsm12 is potentially misleading.

RESPONSES TO REVIEWERS' COMMENTS

Reviewer #1 (Remarks to the Author):

The authors have very carefully considered my points and addressed them fully and in a constructive manner. The new data added is impressive and solidifies the claims made in the previous version. This work is a major advance to the field.

Minor points

1. L55 I think Ref 14 somewhat addresses the ion permeability problem, and could be cited as such.

Ref 14 is already cited above (line 50) as a report supporting Ca²⁺ permeability. Here lists reports that didn't support.

2. L56 IPs of TPCs are associated with high affinity NAADP binding Ref 27

Additional sentence and citation are added here.

3. L195 change "exam" to "examine"

Corrected.

4. L280 change "bindings" to "binding"

Changed.

5. In the discussion the reduced selectivity of NAADP/NADP should be discussed fully in relation to the exquisite selectivity of recombinant Lsm12 for NAADP over NADP.

We added some discussion in the result for the difference.

6. JTP2 has recently been suggested to also act as an NAADP binding protein in 2 papers in Science Signaling a couple of months back. This should be discussed in the context of this (much for convincing and rigorous) study

The identification of JTP2 as another NAADP binding protein is a significant progress and finding. We added discussion and citation of these 2 papers.

Antony Galione

Reviewer #2 (Remarks to the Author):

The authors have significantly improved the manuscript presentation, and addressed all of my concerns and comments from the initial submission. I am therefore satisfied with the quality of data and conclusions the authors have made and presented in this new version. Very promising data and results overall.

Reviewer #3 (Remarks to the Author):

The additional data added in response to the original review has addressed many of the prior concerns. The final study is comprehensive in scope and well executed. Overall this paper will be of exceedingly high impact for the calcium signaling field.

Two comments.

(1) Papers on HN1L /JPT as a candidate NAADP receptor should be cited given their recent publication.

The identification of JPT2 as another NAADP binding protein is a significant progress and finding. We added discussion and citation of these 2 papers.

(2) Figure 3c. The text should state the measured IC50s for NAADP and NADP displacement, consistent with Panels 3a and 3b.

We added the fitted value for Kd of NAADP and NADP in text for Figure 3c.

(3) Unclear about the methods for new Fig 3c which the authors describe as 'HEK293 endogenous LSm12 pulled down by immobilized NAADP'. If this is simply probing HEK293 lysates there is no real support for the statement that this preparation is just LSM12, as indeed the different binding results may suggest. The statement that this preparation represents endogenous Lsm12 is potentially misleading.

We agree that the protein sample pulled down by immobilized NAADP contained not only Lsm12 as it was seen in our quantitative mass spec data. However, we probed with anti-Lsm12 antibody and thus the readout reflects the Lsm12 pulled down by immobilized NAADP. There is possibility that the immobilized NAADP may indirectly pull down Lsm12 by direct pull-down of some Lsm12 interacting protein. However, we consider this to be unlikely given that NAADP can directly bind to purified Lsm12. With caution, we have modified the text by avoid saying the binding between NAADP and endogenous Lsm12.